# Staufen1 links RNA stress granules and autophagy in a model of neurodegeneration

Sharan Paul[1], Warunee Dansithong[1], Karla P. Figueroa[1], Daniel R. Scoles[1] & Stefan M. Pulst[1]

Spinocerebellar ataxia type 2 (SCA) is a neurodegenerative disease caused by expansion of polyglutamine tract in the ATXN2 protein. We identified Staufen1 (STAU1) as an interactor of ATXN2, and showed elevation in cells from SCA2 patients, amyotrophic lateral sclerosis (ALS) patients, and in SCA2 mouse models. We demonstrated recruitment of STAU1 to mutant ATXN2 aggregates in brain tissue from patients with SCA2 human brain and in an SCA2 mouse model, and association of STAU1 elevation with dysregulation of SCA2-related transcript abundances. Targeting *STAU1* in vitro by RNAi restored *PCP2* transcript levels and lowering mutant *ATXN2* also normalized STAU1 levels. Reduction of *Stau1* in vivo improved motor behavior in an SCA2 mouse model, normalized the levels of several SCA2-related proteins, and reduced aggregation of polyglutamine-expanded ATXN2. These findings suggest a function for STAU1 in aberrant RNA metabolism associated with ATXN2 mutation, suggesting STAU1 is a possible novel therapeutic target for SCA2.

[1] Department of Neurology, University of Utah, 175 North Medical Drive East, 5th Floor, Salt Lake City Utah 84132, USA. These authors contributed equally: Sharan Paul, Warunee Dansithong. Correspondence and requests for materials should be addressed to S.M.P. (email: stefan.pulst@hsc.utah.edu)

Spinocerebellar ataxia type 2 (SCA2) is an autosomal dominant cerebellar ataxia characterized by progressive degeneration of cerebellar Purkinje cells (PCs), and selective loss of neurons within the brainstem and spinal cord[1–4]. The mutation in SCA2 is expansion of a CAG repeat in exon-1 of the ATXN2 gene encoding a polyglutamine (polyQ) domain. PolyQ expansions in ATXN2 result in toxic gain of function associated with neuronal protein aggregates[5,6]. ATXN2 aggregates, degeneration of cerebellar PCs and altered RNA expressions are pathological consequences of ATXN2 mutation in SCA2 patients and animal models[7–10].

ATXN2 is widely expressed in the mammalian nervous system, and is involved in diverse cellular functions involving inositol 1,4,5-triphosphate receptor (IP3R) and EGF receptor signaling as well as translation and embryonic development[1,9,11–15]. ATXN2 interacts with multiple RNA-binding proteins (RBPs), including RNA splicing factor A2BP1/Fox1, polyA binding protein 1 (PABP1), DDX6, and Tar DNA-binding protein-43 (TDP-43) demonstrating its unique role in RNA metabolism[15–20]. Furthermore, ATXN2 is a constituent protein of stress granules (SGs) and P-bodies, suggesting its function in sequestering mRNAs and regulating protein translation during stress[16,17,21–23].

The double-stranded RNA-binding proteins, Staufen1 (STAU1) and Staufen2 (STAU2) are recruited to cytoplasmic inclusions in brain oligodendrocytes and cultured cells and modulate SGs dynamics[24,25]. STAU1 is a multifunctional protein involved in regulating RNA metabolism, but also with mRNA transport in neuronal dendrites, and other cells in vertebrates[26–30]. STAU1-deficient mice exhibit defects in dendritic mRNA transport and neuron morphology with reduced synapse formation[31]. STAU1 together with TDP-43 and FMRP is involved in ribonucleoprotein particle transport in neuronal dendrites. Dysregulation of the STAU1/TDP-43/FMRP complex sensitizes neurons to death[32,33]. Furthermore, STAU1 regulates the translational efficiency via 5′ UTR and polysome association, and the stability of specific transcripts through their 3′UTRs, a mechanism referred to as STAU1-mediated RNA decay (SMD)[34–36].

Mutant polyQ proteins have been associated with dysfunction in the ubiquitin-proteosome system (UPS) and the autophagic system. Autophagy dysfunction is associated with many neurodegenerative diseases including amyotrophic lateral sclerosis (ALS), Huntington disease (HD), and Autism spectrum disorders (ASD)[37–39]. Stimulating autophagy is beneficial for HD, frontotemporal degeneration (FTD) with ALS, and ASD disease models[38–40]. However, the role of autophagy dysfunction in SCA2 pathology and its link to dysregulated mRNA levels is poorly understood.

In this study, we show that STAU1 steady-state levels are increased in cells from SCA2 and ALS patients as well as in SCA2 animal models. SGs are increased in SCA2-derived cells even under physiologic conditions and STAU1 is recruited to mutant ATXN2 aggregates in SCA2 fibroblasts. We establish a function for STAU1 in regulating abundance of mRNA transcripts in a manner that mimics the defects observed in SCA2 cellular and animal models. Furthermore, reducing STAU1 levels restored expression of several SCA2-related proteins in vitro and in vivo. We establish a novel role for STAU1 in dysregulated RNA metabolism, and demonstrate that lowering STAU1 expression can restore specific aspects of SCA2 pathology. STAU1 may represent a therapeutic target for certain neurodegenerative diseases.

## Results

### ATXN2 and Staufen1 co-localization and interaction in SCA2.
An association of STAU1 and SGs in brain oligodendrocytes and other cultured cells was previously described[24,25]. Because ATXN2 is a component of SGs[16,17], we investigated if ATXN2 and STAU1 co-localized under conditions of stress. The specificity of anti-Staufen antibody was confirmed by multiple methods (Supplementary Fig. 1a–d). Following exposure of HEK-293 cells to arsenite (oxidative stress), we assessed ATXN2 and STAU1 co-localization by immunofluorescence. Arsenite-induced stress resulted in co-localization of ATXN2 and STAU1 in cytoplasmic SGs positive for TIA-1, a marker for SGs[41] (Fig. 1a).

SCA2 cells are under intracellular stress caused by expression of toxic mutant ATXN2. As ATXN2 aggregates are evident in SCA2 brains[5–8], we investigated the recruitment of STAU1 to ATXN2 aggregates in SCA2 fibroblasts (FBs) under normal and stress conditions. Under physiological conditions (without added stress), SCA2 FBs (ATXN2-Q22/42) demonstrated elevated numbers of SG-like aggregates positive for both ATXN2 and STAU1 that were rarely observed in normal FBs (ATXN2-Q22/22). By ImageJ, the average area per aggregate was about 15% higher for SCA2 cells over wild-type cells at 37 °C (Fig. 1b–d). Under conditions of stress, increased numbers of ATXN2-STAU1 granules were seen in normal and SCA2 FBs, but granule numbers were more pronounced in SCA2 cells (Supplementary Fig. 1e, f). We then investigated Staufen1 in vivo. Double immunostaining for ATXN2 and STAU1 demonstrated that both co-localized to aggregates in cerebellar PCs of $ATXN2^{Q127}$ mouse (Fig. 1e) and individuals with SCA2 (Fig. 1f), but not in unaffected controls. Our observation of strong STAU1 expression in human PCs is consistent with STAU1 in cerebellar PCs reported by the Human Protein Atlas (http://www.proteinatlas.org/).

Co-localization of ATXN2 and STAU1 in SG-like aggregates in SCA2 cells predicts physical interaction between the two proteins. To test this hypothesis, HEK-293 cell extracts (with or without RNase A treatment) expressing Flag-tagged ATXN2-Q22 or ATXN2-Q108 were subjected to immunoprecipitation (IP) with Flag monoclonal antibody (mAb) beads. Western blot analysis showed expression of Flag-ATXN2 proteins as expected, as well as RNA-dependent interaction of endogenous STAU1 with Flag-ATXN2-Q22 or Flag-ATXN2-Q108 (Fig. 1g). Notably, an increased level of STAU1 was seen in input cellular extracts when Flag-ATXN2-Q108 was overexpressed but not Flag or Flag-ATXN2-Q22. The ATXN2 immunoprecipitates also showed co-IP of PABPC1 (RNA-dependent) and DDX6 (both RNA and non-RNA-dependent), both of which are known ATXN2 interactors[9,15–17]. Taken together, these data support physical interaction and co-localization of ATXN2 and STAU1 in SG-like aggregates in SCA2 cells.

### STAU1 levels are increased in neurodegenerative disease.
Because STAU1 interacts with ATXN2 and forms aggregates with mutant ATXN2 in SCA2 cells, we measured steady-state levels of STAU1 in SCA2- FBs and Epstein-Barr virus (EBV) immortalized lymphoblastoid B cells (LBCs). We used SCA2 skin FBs with CAG (35, 35, 35, 42, and 45) repeats and SCA2 LBCs with CAG (40, 46, and 52) repeats in ATXN2. Expression of ATXN2 (CAG repeats) in SCA2- FBs or LBCs was validated by reverse transcription polymerase chain reaction (RT-PCR) analyses (Supplementary Fig. 2a, b). Whole-cell extracts were prepared with Laemmli sodium dodecyl sulfate–polyacrylamide gel electrophoresis (SDS–PAGE) sample buffer[9,42], and analyzed by western blotting to determine STAU1 levels. Compared to normal control cells, SCA2- FBs and LBCs showed increased STAU1 levels with DDX6 protein levels unchanged (Fig. 2a, b). As HD and SCA3 are caused by CAG repeat expansions, HD and SCA3 FBs would be expected to exhibit increased STAU1 levels. Surprisingly, no significant alterations of STAU1 levels were observed in HD and SCA3 FBs (Fig. 2a) or HEK-293 cell expressing mutant ATXN3

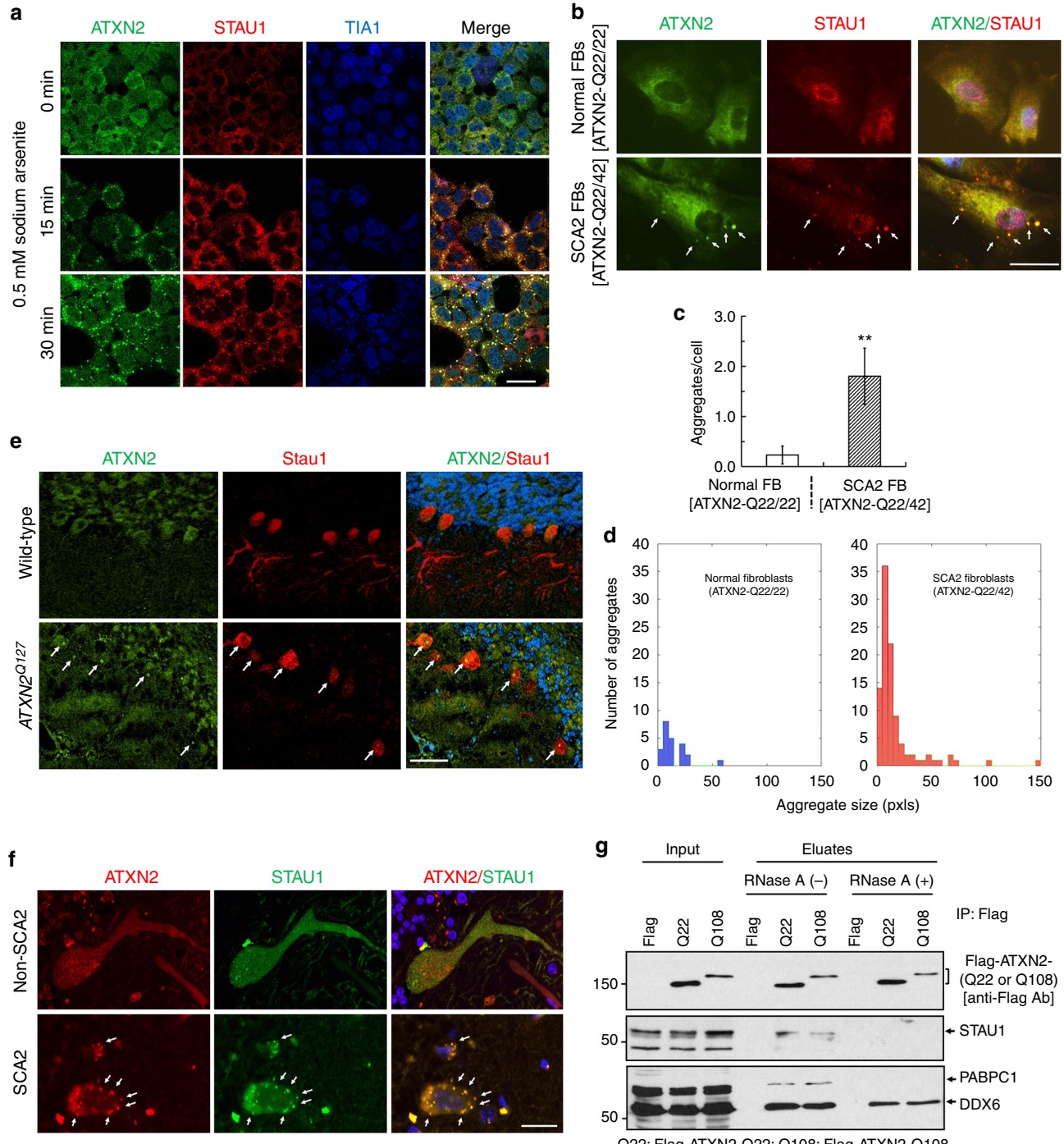

**Fig. 1** Co-localization of Staufen1 with ATXN2 in stress-granule-like structures. **a** ATXN2 and STAU1 co-localize in SGs. Immunostaining of HEK-293 cells with antibodies against ATXN2, STAU1, and TIA-1 show co-localization of STAU1 with ATXN2 and TIA-1 in SGs during stress (0.5 mM sodium arsenite for 15 and 30 min) that are not present in the absence of stress. Scale bar, 10 μM. **b** Constitutively present SG-like structures positive for both ATXN2 and STAU1 in SCA2 FBs, but not in normal FBs (white arrows). Cells were stained with antibodies to ATXN2 and STAU1. Scale bar, 100 μM. **c** Increased numbers of aggregates in SCA2 FBs at 37 °C. Aggregates > 4 pixels per cell positive for both ATXN2 and STAU1 are shown. One-hundred normal and 96 SCA2-FBs were used for analyses. Data are mean ± SD, **$P < 0.01$, Student $t$-test. **d** Histograms representing the quantities and sizes of ATXN2-STAU1 co-localized granules in normal vs. SCA2 FBs. **e** In vivo co-localization of ATXN2 (green) with Stau1 (red) to aggregates in cerebellar PCs of 24-weeks-old $ATXN2^{Q127}$ mice (white arrows). Scale bar, 30 μM. **f** Co-localized ATXN2 (red) and STAU1 (green) to aggregates (white arrows) in cerebellar PCs from a human SCA2 brain, that are absent in an unaffected control. Scale bar, 30 μM. **g** Immunoprecipitation of ATXN2 with STAU1. Non-RNase A or RNase A treated HEK-293 cell extracts expressing Flag-tagged ATXN2-(Q22 or Q108) were subjected to immunoprecipitation with Flag mAb beads and analyzed by western blotting. STAU1 shows RNA-dependent interactions with wild-type and mutant ATXN2. ATXN2 also co-immunoprecipitates with DDX6 and PABPC1, known ATXN2 interactors. Representative blots of three independent experiments are shown

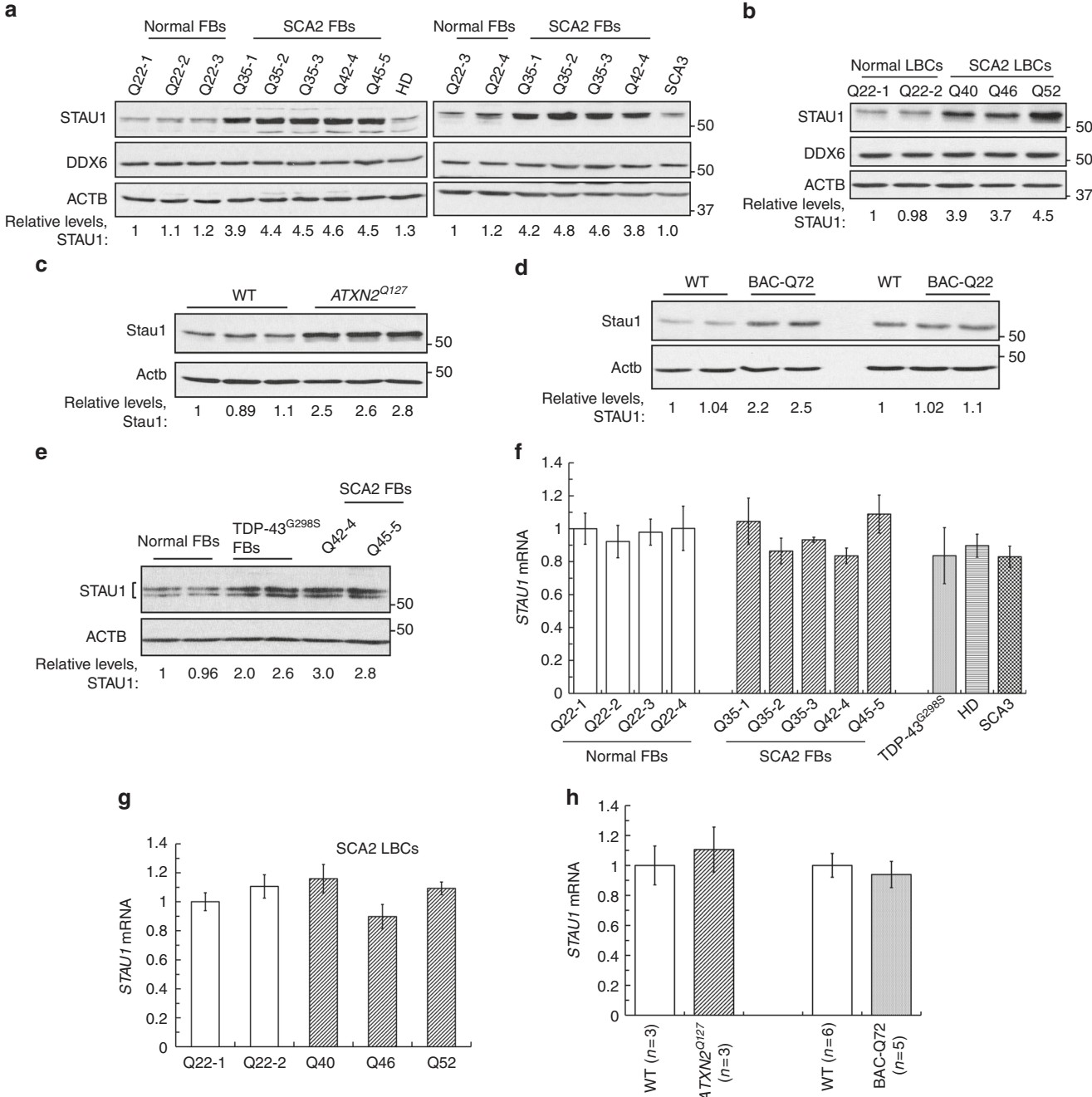

**Fig. 2** Staufen1 protein but not mRNA steady-state levels are increased in neurodegenerative disease cells and tissues. Western blot analysis of SCA2- FBs (**a**) and LBCs (**b**) show increased STAU1 levels compared with normal controls. DDX6 levels are unchanged. HD and SCA3 patient (polyQ expanded) FBs were used as additional controls. Four normal and five SCA2 FBs, and two normal and three SCA2 LBCs were used. **c, d** Western blot analyses of $ATXN2^{Q127}$ (**c**) and BAC-Q72 (**d**) mouse cerebellar extracts (24 weeks of age) showing increased Stau1 levels compared with wild-type or BAC-Q22 controls ($n = 2-3$ animals per group). **e** Western blot of FB extracts from an ALS patient with the TDP-43$^{G298S}$ mutation show increased STAU1 levels. β-Actin was used as loading control and representative blots of three independent experiments are shown. **f–h** STAU1 RNA levels are unaltered in SCA2 and ALS cells and SCA2 mice. qRT-PCR analyses of STAU1 mRNA in SCA2 FBs and ALS FB with TDP-43$^{G298S}$ mutation (**f**) or SCA2 LBCs (**g**). **h** qRT-PCR analyses of cerebellar RNAs from $ATXN2^{Q127}$ and BAC-Q72 mice compared to wild-type littermates (24 weeks of age; $n =$ animals per group). Gene expression levels were normalized to Actb

(Flag-ATXN3-Q56) (Supplementary Fig. 2c). Next, we expanded our investigation to determine if Stau1 levels were abnormally increased in the cerebella of SCA2 mice ($ATXN2^{Q127}$ and BAC-Q72) (refs. [8,9]). Compared to wild-type mice, cerebellar extracts from $ATXN2^{Q127}$ and BAC-Q72 mice (24 weeks of age) had increased abundances of Stau1, but Stau1 increases were not observed in BAC-Q22 mice (Fig. 2c, d). The sensitivity of STAU1

to cellular stress prompted us to examine whether this pathway was active in other neurodegenerative disease conditions. We used ALS-patient FBs harboring the TDP-43$^{G298S}$ mutation and observed increased STAU1 levels compared to control FBs (Fig. 2e). Altogether, these findings establish that STAU1 levels are significantly increased as a consequence of mutations in two different genes leading to neurodegeneration.

Next, we tested whether increased STAU1 was a direct consequence of mutations in ATXN2 or TDP-43. Mutant TDP-43 or aggregations of wild-type TDP-43 are associated with ALS[43,44]. The exogenous expression of ATXN2-Q58 or -Q108 resulted increased STAU1 levels. DDX6 protein levels were unchanged (Supplementary Fig. 2d). Likewise, expression of wild-type or mutant TDP-43 caused increased STAU1 levels as well (Supplementary Fig. 2e).

**Increased STAU1 abundance is caused by altered protein turnover.** To identify the underlying mechanism of increased STAU1 levels, we examined Staufen1 transcript levels in SCA2 and ALS cells, and pathological tissues from SCA2 mouse models. Results of quantitative reverse transcription polymerase chain reaction (qRT-PCR) analyses showed few differences in Staufen1 transcript levels with ATXN2 or TDP-43 mutations (Fig. 2f–h). Given the role of misfolded polyglutamine-expanded proteins in abnormal protein homeostasis, we hypothesized that mutant ATXN2-induced STAU1 abundance might be due to increased stability or impairment of proteasome and/or autophagy pathways. To determine STAU1 stability with ATXN2 mutation, we performed cycloheximide (CHX)-chase experiments demonstrating slower STAU1 degradation in SCA2 FBs compared to normal cells (Fig. 3a, b).

In order to investigate STAU1 metabolism under physiologic gene dosage we used CRISPR/Cas9 genome editing[45] to introduce an expanded CAG58 (Q58) repeat into one ATXN2 allele in human HEK-293 cells. We designated these ATXN2-Q22/58 knock-in (KI) cells. ATXN2-Q22/58 KI cells had elevated STAU1 abundance and ATXN2-STAU1 SGs. PCP2 proteins levels were also reduced in the cellular model, recapitulating expression changes in SCA2 mouse cerebella[8,9] (Fig. 3c and Supplementary Fig. 3). In prior studies, we had established key transcriptomic changes in SCA2 mouse models including reduced abundances in cerebellar PCP2, CACNA1G, and ITPR1 mRNAs[8,9]. These changes were replicated in ATXN2-Q22/58 KI cells in the presence of one mutant ATXN2 allele (Supplementary Fig. 3).

To test if STAU1 is degraded by the proteasome, we assayed STAU1 levels in asynchronous HEK-293 and ATXN2-Q22/58 KI cells treated with the proteasome inhibitor MG132. Compared to untreated cells, the levels of STAU1 were not significantly changed in MG132 treated HEK-293 cells (DMSO vs. MG132 treated groups at each time point) (Fig. 3d, e). Likewise, no significant changes in STAU1 levels were observed in ATXN2-Q22/58 KI cells upon MG132 treatment (DMSO vs. MG132 treated groups at each time point) (Fig. 3d, e) suggesting that STAU1 is not substrate of proteasome pathway. As expected, both HEK-293 and ATXN2-Q22/58 KI cells showed significantly increased levels of Cyclin B1, a proteasome substrate, upon proteasome inhibition.

As STAU1 was not degraded by the proteasome, we hypothesized that abnormal autophagy might explain elevated STAU1 in disease models. Inefficient turnover of light chain 3 isoform II (LC3-II, the lipidated isoform of LC3-I) in conjunction with accumulation of sequestosome 1 (SQSTM1/p62), correlates with autophagy malfunction[46]. On western blots, ATXN2-Q22/58 KI cells and SCA2 FBs showed impaired autophagy, as indicated by the increased levels of LC3-II and p62 (Fig. 3c, f). Accumulation of LC3-II is evidence for efficient autophagosome synthesis prior to degradation at the autolysosome, but also for inefficient autophagosome-lysosome fusion (autolysosome production). To test if LC3-II was increased in SCA2 cells due to inefficient autophagosome-lysosome fusion, we assayed LC3-II in the presence of bafilomycin A1 (Baf, autophagosome-lysosome

fusion inhibitor)[47,48]. Dose-wise Baf treatment resulted in increased LC3-II and p62 levels in HEK-293 and ATXN2-Q22/58 KI cells Fig. 3g). We also observed significant increases of STAU1 in HEK-293 and ATXN2-Q22/58 KI cells upon Baf treatment (Fig. 3g, h) suggesting inefficient STAU1 clearance. Thus, these data suggest that aberrant protein homeostasis in SCA2 cells occurs through mutant ATXN2 and impaired autophagy.

**STAU1 and aberrant mRNA processing.** Previously, we established transcriptome and protein profiles characterized by progressive changes in the course of degeneration in SCA2 mouse models[8,9]. Two proteins that are progressively reduced in SCA2 mouse models are Pcp2 and Calb1, both of which are highly expressed in PCs. In SCA2 FBs and SCA2 LBCs, Pcp2 mRNAs are significantly reduced in the presence of mutant ATXN2 (Fig. 4a, b). As neither PCP2 or CALB1 protein are translated in FBs or LBCs, we examined the effect on PCP2 and CALB1 in HEK-293 cells (Fig. 4c, d). When STAU1 was overexpressed, we observed reduced PCP2 protein, as well as reduced PCP2 and CALB1 mRNA abundances. Endogenous ATXN2 mRNA levels were measured to exclude selective cellular toxicity of STAU1 over-expression, and were found to be invariant (Fig. 4d). Thus, these data demonstrate that aberrant mRNA levels in vivo can be in part recapitulated in a cell culture model.

We next analyzed STAU1 function in the regulation of PCP2 mRNA stability in cultured cells. MYC-tagged PCP2 cDNA (including 5′ and 3′ UTRs, or excluding the 3′UTR) was overexpressed in short-term hygromycin selected HEK-293 cells expressing Flag-tagged STAU1. It is known that STAU1 can bind to 5′ or 3′ UTRs of specific mRNAs to regulate their expression[34–36]. We tested the requirement for the 5′ or 3′ UTRs by cloning MYC-tagged PCP2 mRNAs with these mRNA segments, and determined protein levels by western blotting. Exogenous Flag-STAU1 led to reduction of PCP2 only in the presence of the 3′UTR (Fig. 4e–h). We further tested the importance of the 3′UTR by transient co-expression of Flag-tagged STAU1 with Luciferase-PCP2-3′UTR in HEK-293 cells (Fig. 4i). STAU1 regulates PCP2 expression via its 3′ UTR consistent with SMD.

We next proceeded to test direct interaction between STAU1 and the PCP2 3′UTR. Non-RNase A treated HEK-293 cell extracts overexpressing Flag-tagged STAU1 were subjected to protein–RNA IP with Flag mAb beads, and eluted protein–RNA complexes were divided into two aliquots for western blot and RT-PCR analyses. Flag-STAU1 showed co-IP of ATXN2 (Fig. 5a). RT-PCR analyses from the second aliquot revealed that Flag-STAU1 pulled down PCP2 but not control GAPDH mRNA (Fig. 5a). CALB1 mRNA was also pulled down but was not further studied for its direct interaction with STAU1.

To confirm direct binding between STAU1 and its targets, northwestern blotting experiments were performed using bacterially expressed recombinant His-tagged STAU1 protein and in vitro transcribed DIG-labeled PCP2 probes. We generated several probes: DIG- PCP2[(5′ + 3′)UTR] (b), DIG-PCP2(3′ UTR) (c), and DIG-PCP2(5′UTR) (d) (Fig. 5), which allowed us to identify those parts of the PCP2 RNA that interacted with STAU1. STAU1 bound to PCP2[(5′ + 3′)UTR] and PCP2(3′ UTR) RNAs but not to PCP2(5′UTR) RNA, thereby showing the specificity of the interaction (Fig. 5e). Altogether, these results demonstrate that STAU1 interacts with the PCP2 mRNA via its 3′ UTR.

To test whether the direct interaction occurs in SGs, we performed fluorescent in situ hybridization (FISH) using a PCP2-Cy3 probe. We showed that PCP2 mRNA co-localized with STAU1 aggregates (Fig. 5f).

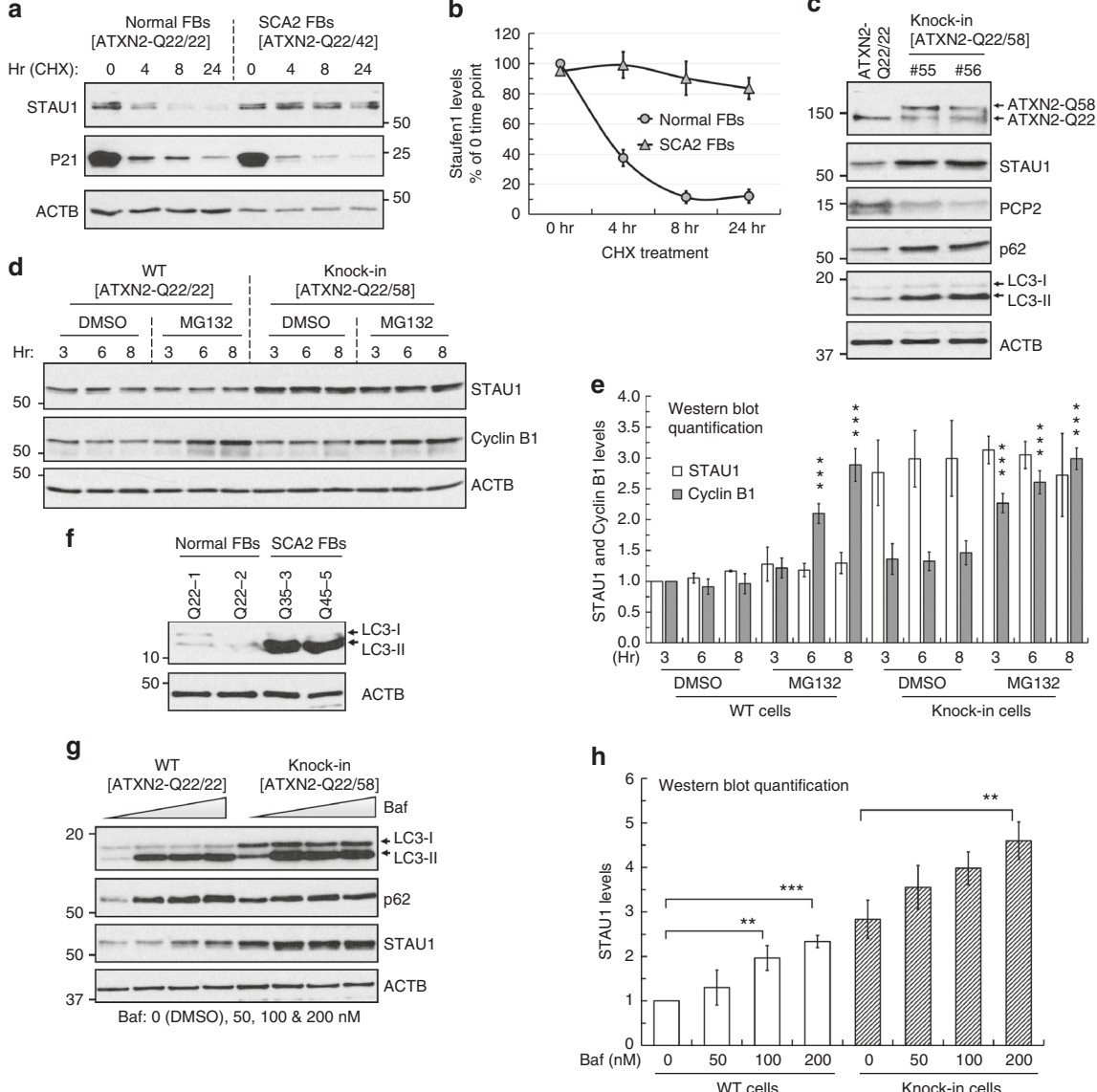

**Fig. 3** Increased STAU1 abundance caused by abnormal autophagy. **a**, **b** Stability of STAU1 in SCA2 FBs. Normal and SCA2 FBs were exposed to CHX (25 µg/ml) for different times from 0 to 24 h, and protein extracts were immunoblotted for STAU1. STAU1 levels are quantified as a percentage of the value for the 0 time point and p21 protein levels are shown to verify CHX inhibition. STAU1 protein stability was prolonged in SCA2 FBs compared to normal FBs. **b** Quantification of STAU1 on western blots is shown. **c** ATXN2-Q22/58 KI cells show increased STAU1, p62 and LC3-II levels on western blots. PCP2 level is decreased in this cell model. **d**, **e** STAU1 is not degraded by the proteasome pathway. **d** HEK-293 and ATXN2-Q22/58 KI cells were treated with proteasome inhibitor MG132 (10 µM) or DMSO as control at different time points and cell extracts were analyzed by western blotting. HEK-293 and ATXN2-Q22/58 KI cells show insignificant changes in STAU1 levels compared to controls upon proteasome inhibition. Increased Cyclin B1 protein levels indicate successful proteasome pathway inhibition. **e** Quantification of STAU1 and Cyclin B1 on western blots. Two-way ANOVA followed by Tukey's multiple comparison test. Data are mean ± SD, ns = $P > 0.05$, ***$P < 0.001$: DMSO vs. MG132 treated groups (6 and 8 h for HEK-293 cells, and each time points for ATXN2-Q22/58 KI cells). **f**–**h** Inefficient autophagy in the presence of mutant ATXN2. **f** SCA2 FBs show inefficient autophagosome-lysosome fusion as indicated by increased LC3-II levels on western blots. **g**, **h** Bafilomycin A1 (Baf) lowers STAU1 clearance through inefficient autophagosome-lysosome fusion. **g** HEK-293 and ATXN2-Q22/58 KI cells were treated dose-wise with Baf for 6 h and analyzed by western blotting. Baf treatment showing dose-wise increased levels of LC3-II and p62 levels and significant increases in STAU1 levels for both cell types compared to untreated cells. **h** Quantification of STAU1 on western blots. One-way ANOVA followed by Tukey's multiple comparison test. Data are mean ± SD, ns = $P > 0.05$, **$P < 0.01$, ***$P < 0.001$. β-Actin was used as loading control. Representative blots of three independent experiments are shown

**Reduced Staufen1 levels ameliorate SCA2 phenotypes**. The increased abundance of STAU1 in neurodegenerative diseases and its association with dysregulation of RNA processing in SCA2 led us to predict that lowering STAU1 levels might be beneficial. In qRT-PCR analyses, STAU1 depleted SCA2 (ATXN2-Q22/52) LBCs by RNAi demonstrated restoration of *PCP2* mRNA expression similar to that in normal LBCs treated with control RNAi without affecting ATXN2 transcript levels (Fig. 6a–c). Consistent with this, we also observed that dose-wise depletion of STAU1 had no effect on *ATXN2* protein levels in control or ATXN2-Q22/58 KI cells (Supplementary Fig. 4a). Thus, reducing STAU1 levels rescued the effect of mutant ATXN2 on aberrant *PCP2* mRNA metabolism observed in SCA2.

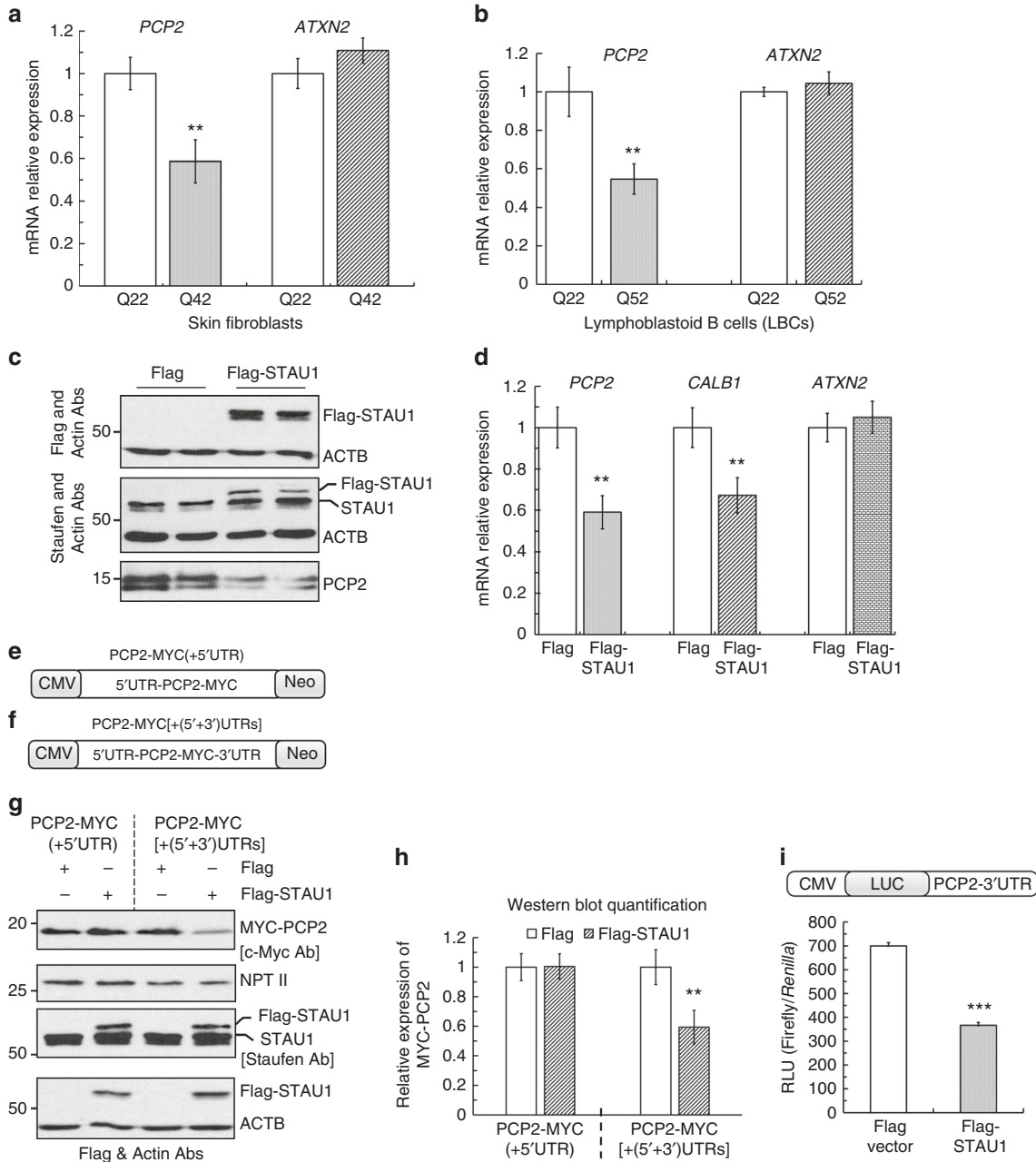

**Fig. 4** Elevated STAU1 levels induce aberrant processing of RNA targets. **a**, **b** In qRT-PCR analyses, SCA2- FBs (**a**) or LBCs (**b**) revealed significantly reduced *PCP2* mRNA abundance compared to normal controls; *ATXN2* mRNA levels are not changed. **c**, **d** Exogenous STAU1 reduces levels of *PCP2* and *CALB1 mRNAs*, two mRNAs greatly reduced in SCA2 mouse models[8, 9]. **c** HEK-293 cells overexpressing Flag-tagged STAU1 were analyzed 48 h post-transfection by western blotting and had reduced PCP2 levels compared to controls. **d** Matching cultures were analyzed by qRT-PCR demonstrating decreased mRNAs for *PCP2* and *CALB1* but not *ATXN2* compared with controls. Gene expression levels were normalized to *GAPDH*. **e–i** Increased STAU1 expression results in reduced PCP2 protein levels in a manner requiring the *PCP2*-3′UTR. Schematics of MYC-tagged *PCP2* cDNA constructs without (**e**) or with (**f**) the 3′UTR. **g** *PCP2* constructs (**e**, **f**) were transfected into short-term hygromycin selected HEK-293 cells expressing Flag-STAU1. Forty-eight hours post-transfection, western blotting of cellular extracts showed significantly reduced PCP2 levels only from the constructs retaining the 3′UTR (**f**). Transfection equalities were monitored by anti-neomycin phosphotransferase (NPTII), expressed independently by the PCP2 plasmids. **h** Quantification of MYC-tagged PCP2 expression on western blots. β-Actin was used as loading control and representative blots of three independent experiments are shown. **i** In the presence of exogenous STAU1, the 3′UTR is required to reduce luciferase expression. Luciferase expression was reported relative to *Renilla* luciferase (RLU). Data are mean ± SD, **$P < 0.01$, ***$P < 0.001$, Student t-test

To test whether depleting mutant ATXN2 in SCA2 cells had an effect on STAU1 abundance, we depleted mutant ATXN2 dose-wise in SCA2- LBCs (ATXN2-Q22/52) or FBs (ATXN2-Q22/42) by *ATXN2* RNAi. Reducing ATXN2 levels demonstrated significant reduction of STAU1 and a significant increase of *PCP2* mRNA levels compared to control RNAi (Fig. 6d–i). In parallel, depletion of ATXN2 by RNAi also resulted in normalization of STAU1 levels in ATXN2-Q22/58 KI cells compared to control RNAi (Supplementary Fig. 4b). Notably, reducing ATXN2 levels did not affect Staufen1 protein and transcript levels in control HEK-293 cells, and in cerebella from *Atxn2* knock-out mice[49] (8 weeks of age) (Supplementary Fig. 4b, c).

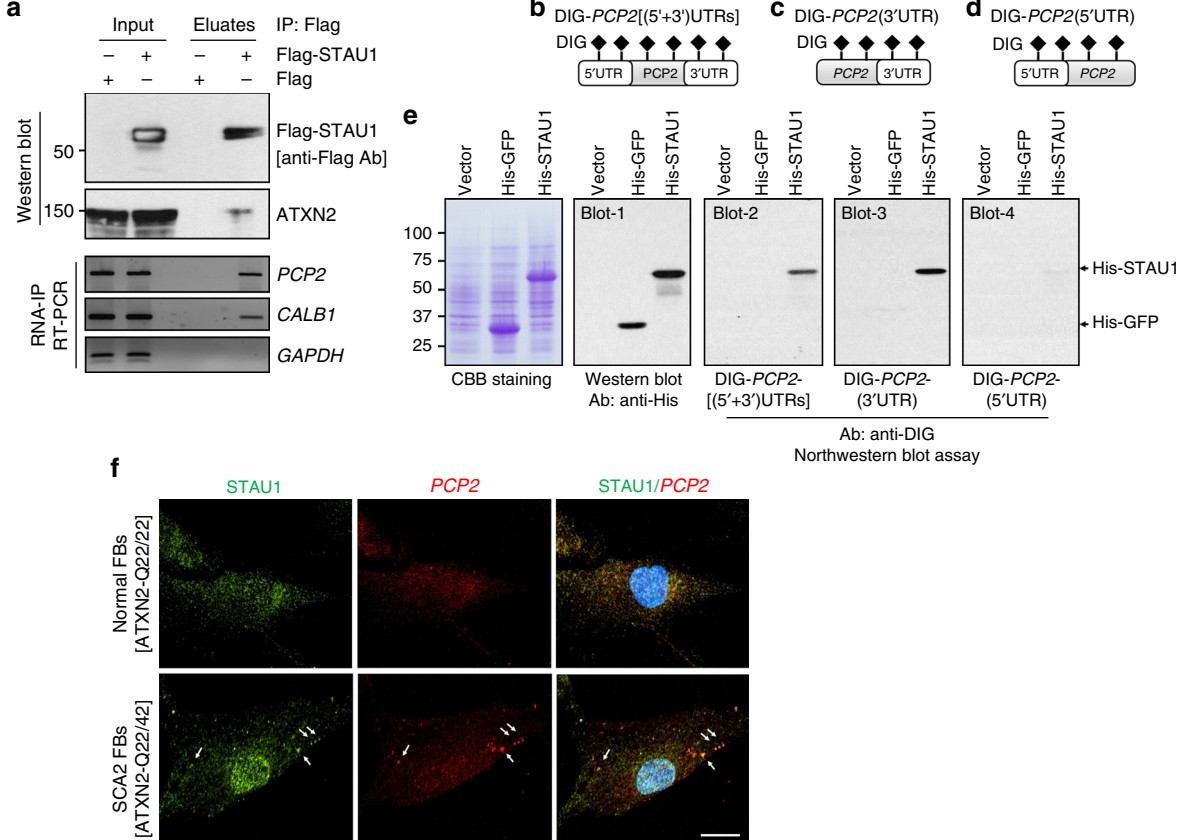

**Fig. 5** STAU1 directly interacts with human *PCP2* mRNA. **a** Non-RNase A treated HEK-293 whole-cell extracts expressing Flag or Flag-STAU1 were subjected to immunoprecipitation with Flag mAb beads. Flag-STAU1 pulled down ATXN2 protein, and *PCP2* and *CALB1* mRNAs on western blot and RT-PCR analysis. **b**–**e** Northwestern blotting showing that STAU1 binds directly to human *PCP2* RNA in a manner requiring the 3′ UTR. Construction of DIG-labeled human *PCP2* RNA probes with both UTRs (**b**), 3′ UTR only (**c**), and 5′UTR only (**d**). **e** Bacterial expressed His-STAU1, His-GFP, or control bacterial lysate (vector alone) were run on SDS–PAGE and stained with Coomassie brilliant blue (CBB) or western blotted with anti-His antibody (blot 1). Protein blots were hybridized with DIG-labeled *PCP2* RNA probes followed by anti-DIG antibody staining. STAU1 directly interacts with *PCP2* RNAs: (5′ + 3′)UTRs (blot 2) or 3′UTR (blot 3), but not with *PCP2(5′UTR)* RNA (blot 4). His-GFP shows no interactions. The blot represents one of three independent experiments. **f** STAU1-*PCP2* RNA granules in SCA2 cells. Normal and SCA2 FBs were immunostained with STAU1 antibodies and STAU1 granules (green) are visualized in SCA2 FBs but not in normal FBs. *PCP2* RNA granules (red) in SCA2 FBs were detected by FISH using a *PCP2*-Cy3 probe. Merged images demonstrate positive STAU1-*PCP2* RNA granules (white arrows) for SCA2 FBs. Scale bar, 30 μM

To test targeting *Stau1* in vivo, we used a well characterized model of cerebellar neurodegeneration expressing mutant $ATXN2^{Q127}$ in PCs[8]. $ATXN2^{Q127}$ mice develop progressive behavioral and proteomic deficits with an onset at 8 weeks. We crossed these mice with a previously generated mouse line deficient for one *Stau1* allele[8,31]. This cross generated four genotypes and all comparisons were made between littermates. Reducing Stau1 dosage by 50% in wild-type animals did not affect rotarod behavior. On the other hand, Stau1 reduction in $ATXN2^{Q127}$ mice lead to an improved performance on the accelerating rotarod (Fig. 7a).

Significant and progressive decreases in key PC mRNAs and even greater decreases in the respective encoded proteins (Calb1, Pcp2, Rgs8, Pcp4, Homer3, and Fam107b) have been described in SCA2 mice[8,9,50]. *Stau1* haploinsufficiency in $ATXN2^{Q127}$ mice significantly increased levels of these six proteins toward normalization (Fig. 7b, c). SCA2 cells with mutant ATXN2 form ATXN2-STAU1 aggregates (Fig. 1). We, therefore, tested whether Stau1 reduction would reduce aggregates in vivo. Whereas aggregates positive for Stau1 and ATXN2 are abundant in PCs of $ATXN2^{Q127}$ mice, *Stau1* haploinsufficiency was associated with virtual absence of aggregates (Fig. 7d). Thus, *Stau1* haploinsufficiency partially restored behavioral,

proteomic, and morphological characteristics of SCA2 pathology in vivo.

## Discussion

Processes regulating RNA metabolism, including translation, transport, storage, and degradation, have important roles in cellular function both in health and diseases. SGs are critical for normal RNA metabolism as they regulate translation during cellular stress. RNA granules can regulate mRNA transport and exert local expression control, and SGs are regulated by autophagy.

Linkages between SGs and neurodegenerative disorders have been well documented. Many protein components of RNA granules are recruited to and modulate formation of SGs, including ATXN2, TDP-43, fused in sarcoma (FUS), survival of motor neuron (SMN) and fragile X mental retardation protein (FMRP)[51]. Misregulation of these RBPs by mutation or other means contributes to neuronal dysfunctions and is a primary cause of multiple neurodegenerative disorders. Gene silencing of *ATXN2* has shown promise in preclinical evaluations using mouse models of SCA2 and ALS[50,52]. ATXN2 regulates RNA metabolism through interacting with RBPs and is localized in

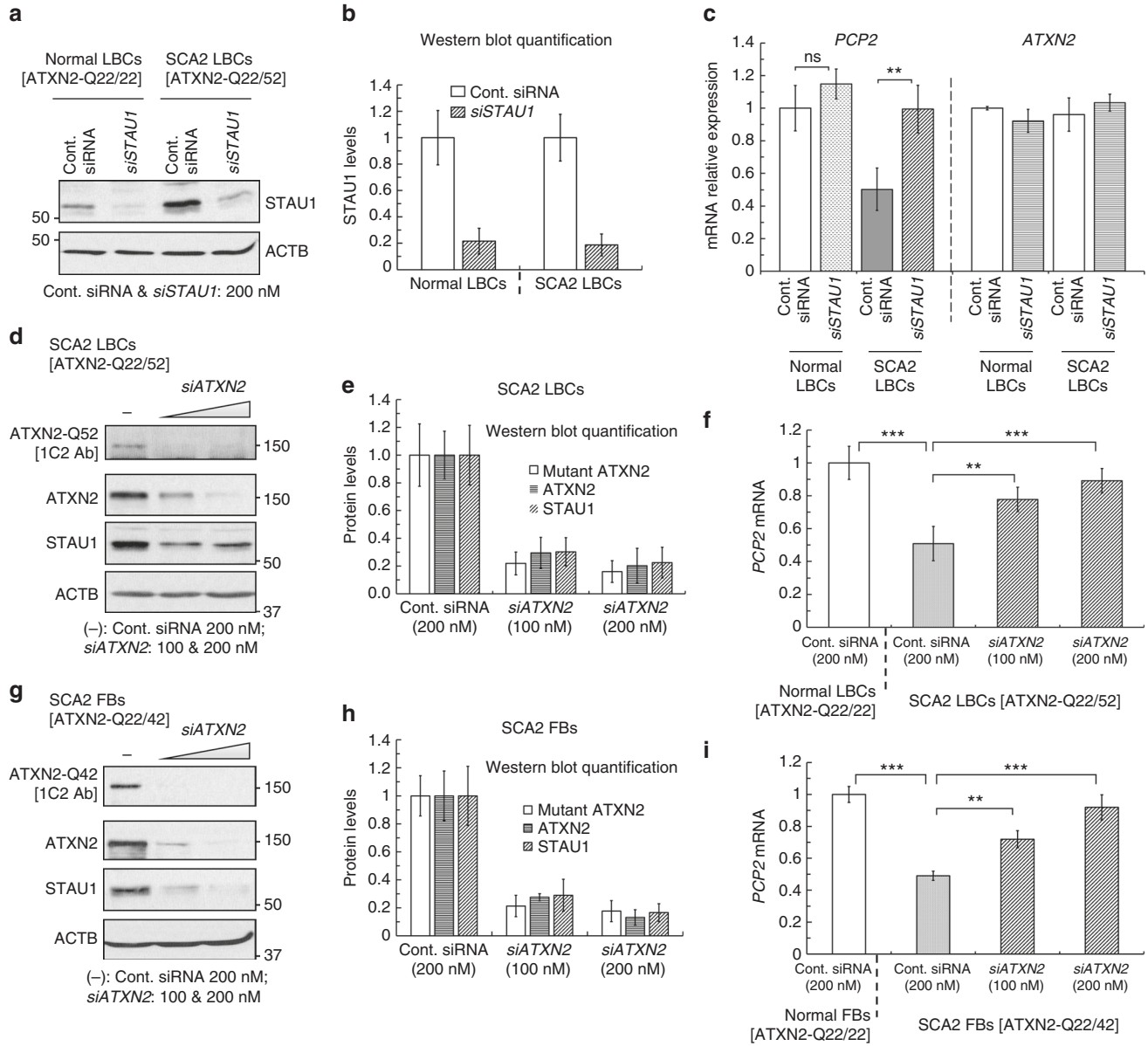

**Fig. 6** Lowering STAU1 restores *PCP2* expression in SCA2 cells. **a** Normal [ATXN2-Q22/22] and SCA2 [ATXN2-Q22/52] LBCs were electroporated with *STAU1* siRNA and STAU1 levels were determined by western blotting 4 days post-electroporation. **b** Quantification of three independent experiments of the blot in **a**. **c** Matched cultures were used for qRT-PCR analyses to measure *PCP2* transcript levels. STAU1 depletion is associated with increased *PCP2* transcript levels in SCA2 LBCs. *PCP2* mRNA is unchanged upon STAU1 knockdown in normal LBCs. STAU1 silencing does not alter *ATXN2* transcript steady-state levels in normal or SCA2 LBCs. **d–i** *ATXN2* siRNA lowers STAU1 levels and increases *PCP2* expression in SCA2 cells. **d** SCA2 LBCs [ATXN2-Q22/52] were electroporated with *siATXN2* and evaluated by western blotting. **e** Quantification of three independent experiments of the blot in **d**. **f** Matched cultures were used for qRT-PCR analyses to measure *PCP2* transcript levels, demonstrating *PCP2* expression increased with *siATXN2* dose. **g–i** As for **d–f** but using SCA2 FBs [ATXN2-Q22/42]. β-Actin and *GAPDH* RNA were used as internal controls for western blot and qRT-PCR analyses, respectively. Abundances relative to β-Actin determined densitometrically. Data are mean ± SD, ns = $P > 0.05$, **$P < 0.01$, ***$P < 0.001$, Student *t*-test

SGs[9,15–20]. It remained unclear, however, whether these interactions and SG-localization were significant in SCA2 pathogenesis in part because no protein binding or SG-localization differences were ever observed between wild-type and mutant ATXN2.

Here, we show for the first time that STAU1 is a novel RNA-dependent interactor for ATXN2 (Fig. 1). The strength of protein–protein interaction did not differ for wild-type and mutant ATXN2, and we found co-localization of STAU1 with ATXN2 aggregates in individuals with SCA2 and in an SCA2 mouse model (Fig. 1). We showed increased STAU1 levels in SCA2- FBs and LBCs, SCA2 mouse cerebellar extracts, and in ALS cells expressing mutant TDP-43 (Fig. 2). We also modeled

SCA2 in cultured cells by using CRISPR/Cas9-mediated gene editing to introduce a CAG repeat expansion in *ATXN2*, and found that these cells have increased steady-state STAU1 levels and a molecular signature partially shared with SCA2 mouse cerebella, including impaired autophagy read-out (increased LC3-II levels) (Fig. 3). These observations suggest that STAU1 elevation is a common feature of some neurodegenerative states.

Staufen was first described as a protein in the fly zygote that was essential for polarized localization of specific mRNAs such as *oskar* and *bicoid*[53,54]. Since flies with a complete absence of Staufen were viable, but could not generate fertile offspring, genes

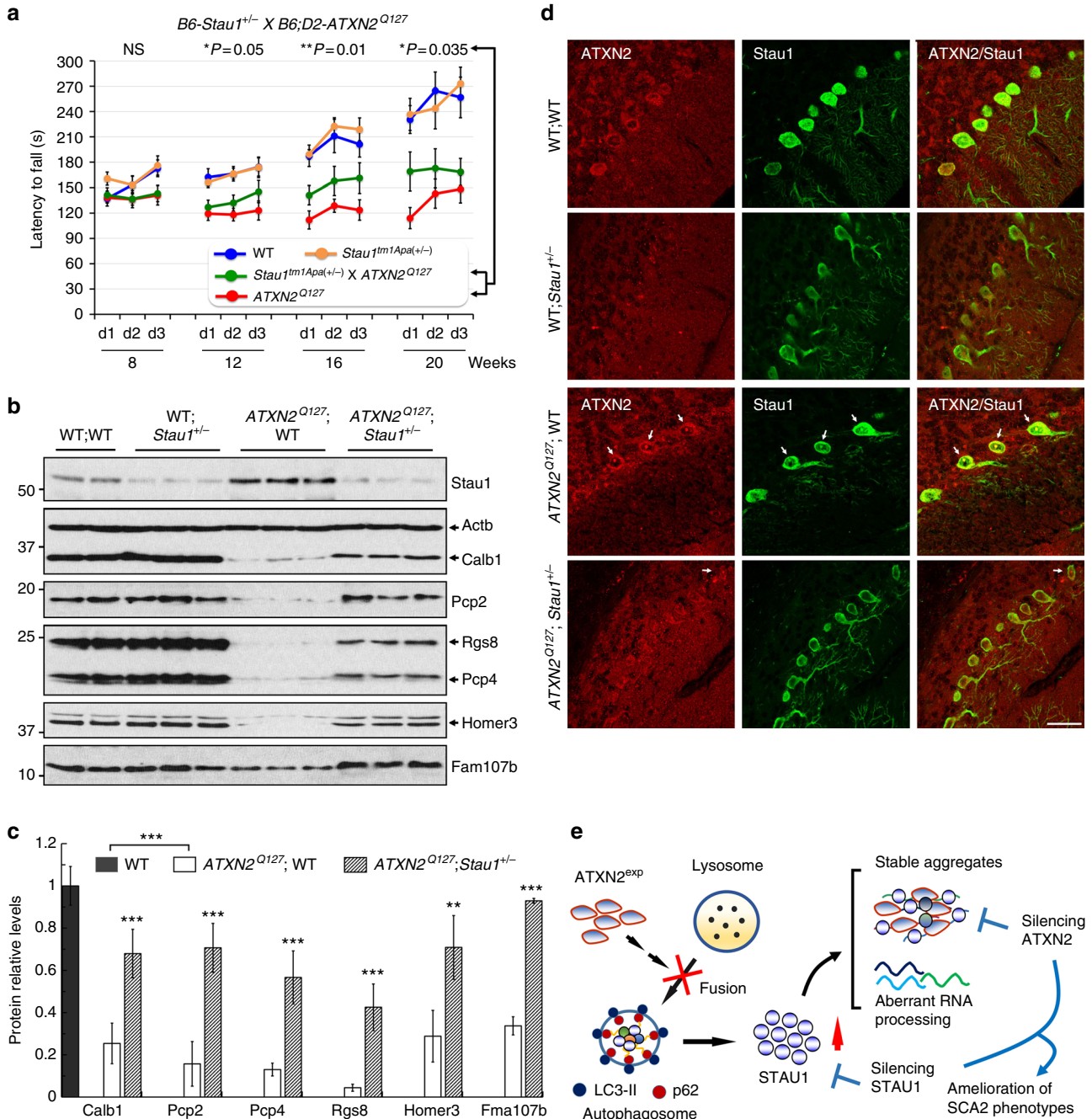

**Fig. 7** Silencing of STAU1 mitigates SCA2 phenotypes. **a** *Stau1* haploinsufficiency improves abnormal motor behavior of *ATXN2*[Q127] mice as determined by rotarod behavior at 8, 12, 16, and 20 weeks of age. *ATXN2*[Q127];*Stau1*[+/−] mice (green) have improved rotarod performance compared with *ATXN2*[Q127] littermates (red) starting at 12 weeks of age. Note that *Stau1* haploinsufficiency (orange) by itself does not alter motor function; $n = 9$–15 mice per group. Values shown are mean ± SE. Significance was determined using generalized estimating equations (GEE). NS, nonsignificant, *$P < 0.05$, **$P < 0.01$. **b**, **c** Reduction of Stau1 in vivo improves levels of key cerebellar proteins towards normalization. **b** Western blotting of cerebellar extracts from *ATXN2*[Q127]; *Stau1*[+/−] mice showing improvement of protein levels for Calb1, Pcp2, Rgs8, Pcp4, Homer3, and Fam107b towards normalization. Each lane represents cerebellar extract from an individual mouse. β-Actin is used as a loading control and the blots are from three replicate experiments. **c** Quantitative analysis of western blots shown in **b**. Data are mean ± SD, **$P < 0.01$, ***$P < 0.001$, Student *t*-test. **d** Combined immunostaining of ATXN2 (red) and Stau1 (green) of cerebellar sections from *ATXN2*[Q127] and crossed *ATXN2*[Q127];*Stau1*[+/−] mice (34 weeks of age) demonstrating reduced ATXN2-Stau1 aggregates in crossed *ATXN2*[Q127];*Stau1*[+/−] mice. Scale bar, 30 μM. **e** Model for STAU1 in the pathology of SCA2 and other neurodegenerative diseases

in this class were designated "grandchildless" and given names of European extinct noble lines (staufen, tudor, valois, vasa, etc.). Subsequently, Staufen was identified in mammalian cells, localized to RNA granules under cellular stress, and implicated in a number of RNA-related functions such as mRNA transport and degradation.

Consistent with the established role for STAU1 in mRNA transport from fly studies, we observed increases in SG numbers in SCA2 FBs (Fig. 1). As steady-state STAU1 levels were consistently increased in multiple SCA2 cell lines, we examined the presence of STAU1-labeled RNA granules under normal vs. stressed conditions. During stress, the number of RNA granules

in SCA2 cells was greater than in normal control cells (Supplementary Fig. 1e, f). The observation of differences requires precise control of the timing of stress due to the dynamic nature of stress-granule assembly[55], which might explain why this was not observed previously[17].

Our data suggest that ATXN2 mutation results in abnormal autophagy leading to increased STAU1 abundance. Autophagy dysfunctions have also been seen in association with other polyglutamine disease genes[38,56,57]. It is also possible that STAU1 is more directly involved in autophagic feedback regulation on ATXN2. Additional work will be required to characterize the role of ATXN2 in abnormal autophagy and the potential function of STAU1 in signaling to the autophagosome.

One of the functions of STAU1 is targeting specific mRNAs for degradation via SMD, a mechanism similar to nonsense-mediated decay[35,36]. To verify this for SCA2 genes, we evaluated expression of PC-specific transcripts that were significantly downregulated in the cerebella of ATXN2[Q127] and BAC-Q72 mice[8,9,50]. PCP2, CALB1, RGS8, PCP4, Fam107b, and Homer3 are highly expressed in mature Purkinje cells and have reduced expression in SCA2 mice, and their expression was improved in ATXN2[Q127] mice haploinsufficient for Stau1 (Fig. 7). We showed that one of these mRNAs, PCP2, precipitated with STAU1 dependent on the presence of its 3′UTR (Fig. 5), consistent with SMD mediated by STAU1 interaction with 3′UTRs[35,36]. STAU1 is known to interact with target RNAs by recognizing double-stranded RNA structures through one or more double-stranded RNA-binding domains[58,59]. Double-stranded RNAs can be formed by intramolecular folding of the 3′UTR upon itself or by heteromolecular interaction, for example with lincRNAs[60]. With a recent demonstration for directly differentiating iPSCs into human PCs[61], it may now be possible to define STAU1-dependent mRNA networks in PCs, in addition to using the extant transcriptome profiles in SCA2 mouse models[8,9].

Although SMD-networks have been identified for several cell types[35,36,62,63], little is known regarding SMD in PCs. Increased SMD in ATXN2[Q127] or BAC-Q72 mice represented an attractive explanation for a large number of downregulated mRNAs in cerebellar transcriptomes at time points preceding any dendritic or cellular loss[8,9]. Some of these RNAs (~ 15.6%) had also been identified as STAU1 interactors by hiCLIP[63] (Supplementary Fig. 5 and Supplementary Table 1), supporting the notion of STAU1-mediated RNA processing in SCA2. The VENN diagram compares two disparate datasets (HEK-293 cells vs. mouse cerebella). Despite this, 176 PC mRNAs are shared as potential direct STAU1 targets. Included among these is Fam107b that is highly reduce in SCA2 mice[8,9,50] (Supplementary Fig. 5 and Supplementary Table 1). The demonstration of STAU1 interaction with PCP2 mRNA in SCA2 SGs labeling positive for both (Fig. 5) supports the hypothesis of STAU1-mediated dysregulation of RNAs in neurodegeneration.

Our observations in SCA2 models of aberrant STAU1 abundances or aggregates and reduced PCP2 mRNA point to STAU1 as a therapeutic target for SCA2. We used RNAi to reduce STAU1 expression in SCA2 cells and were able to restore PCP2 mRNA abundance (Fig. 6). Likewise, reducing mutant ATXN2 in SCA2 cells leads to the same result placing ATXN2 likely upstream of STAU1-mediated effects on PCP2 (Fig. 6). In addition, STAU1 elevation in cultured ALS cells (Fig. 2 and Supplementary Fig. 2e), supports that STAU1 could be a therapeutic target for multiple neurodegenerative diseases.

The significance of STAU1 to neurodegeneration was further emphasized by in vivo studies. We used genetic interaction to reduce Stau1 gene dosage in ATXN2[Q127] mice demonstrating improved motor deficit function, restored expression of cerebellar SCA2-related proteins[8,9,50] (Fig. 7). It is noteworthy that the

levels of SCA2-related proteins were unchanged in wild-type littermates that were haploinsufficient for Stau1, and that the rotarod performance of these mice was normal. Thus, it is noteworthy that significant transcriptomic changes caused by loss of motor neuron autophagy were seen in SOD1 mice but not normal control mice[64]. We also observed that reduced STAU1 levels decreased the burden of ATXN2[exp] aggregates (Fig. 7), which provides mechanistic evidence that STAU1 elevation may potentiate ATXN2[exp]-induced toxicity. Furthermore, interaction between ATXN2 and STAU1 and STAU1-mediated RNA processing (Figs. 1 and 4) support that notion. Our data suggest that STAU1 as a sensor of cellular stress that once engaged, may trigger a rapid response leading to reciprocal effects on autophagy. In the absence of stress, changes of STAU1 levels appear to have relatively minor effects at least within the parameters measured in this study. Beyond ATXN2, strategies that reduce STAU1 abundances could be pursued as a way to mitigate ATXN2[exp] pathological aggregation and neurodegeneration.

Overall, our results demonstrate that ATXN2[exp] expression and consequent elevation of STAU1 level in SCA2 are dually deleterious affecting autophagy function via inefficient autophagosome-lysosome fusion and mRNA stability via SMD, as summarized in Fig. 7e. Lowering STAU1 expression may be a potent therapeutic approach for SCA2 that may be effective for ALS as well.

Misregulated processing of mRNAs has been implicated in a number of neurological diseases[65]. Targeting ribonucleoprotein particle (RNP) assembly in human neurodegenerative diseases is a relatively new approach. Protter and Parker (2016) divided diseases in hypo- and hyper-assembly diseases with spinal muscular atrophy (SMA) falling into the former and ALS into the latter category[65]. SCA2 may be the first known polyQ disease that —at least in part—can now be considered an RNA granule hyper-assembly disease. A central role for STAU1 in the pathogenesis of a neurodegenerative disease has hitherto not been described and should be examined in other diseases. Our data suggest protein dyshomeostasis as a common underlying feature of neurodegenerative diseases and that targeting STAU1 may have broad therapeutic potential. STAU1 may offer unique targets for small molecules and antisense oligonucleotides. Rather than targeting STAU1 expression levels itself it may also be possible to target interaction of STAU1 with specific RNA targets.

## Methods

**DNA constructs**. Human cDNA sequences for STAU1 (NM_001037328) were derived from the NCBI DNA database and used to design primers to PCR-amplify the coding sequences from a cDNA library made from human embryonic kidney 293 (HEK-293) cells. All cDNAs including ATXN2 with expanded CAG repeats[7] were subsequently cloned into the appropriate vectors; pcDNA3.1/Flag (Agilent Technologies, USA) plasmids. For in vitro RNA-binding assay, human STAU1 and GFP coding sequences were PCR amplified and cloned into pET-His plasmids. All constructs were verified by sequencing.

**siRNAs and reagents**. The siRNAs used in this study were: All Star Negative Control siRNA (Qiagen, Cat# 1027280), human siATXN2 [Hs_ATXN2_2 (Qiagen, Cat# SI00308196)], human siStau1: 5′-CCAUAACUACAACAUGAGdTdT-3′ and mouse siStau1: 5′-CAACUGUACUACCUUUCCAdTdT-3′ (ref. [35]). Staufen1 siRNA oligonucleotides were synthesized by Invitrogen, USA. The oligonucleotides were deprotected and the complementary strands were annealed. Cycloheximide (Tocris USA, Cat# 0970), Bafilomycin A1 (InvivoGen USA, Cat# tlrl-baf1) and (S)-MG132 (Cayman Chemical USA, Cat# 10012628) were used in this study.

**Cell line authentication**. In order to adhere with the NIH guideline on scientific rigor in conducting biomedical research (NOT-OD-15-103) on the use of biological and/or chemical resources we have authenticated our cell lines utilizing STR analysis on 24 loci including Amelogenin for sex identification. The kit utilized for this was GenePrint 24 system. https://www.promega.com/products/cell-authentication-sample-identification/mixed-sample-analysis/geneprint-24-system/?catNum = B1870

**Cell culture and transfections**. Four normal skin fibroblasts (ATXN2 with CAG22) and five SCA2 patient-derived skin FBs (ATXN2 with CAG repeats; 35, 35, 35, 42, and 45), were maintained in Dulbecco's Modified Eagle's (DMEM) medium containing 10% fetal bovine serum. Two normal (ATXN2 with CAG22) and three SCA2 patient-derived (ATXN2 with CAG repeats; 40, 46, and 52) Epstein-Barr virus (EBV)-immortalized LBCs were maintained in RPMI medium containing 15% fetal bovine serum. All subjects gave written consent and all procedures were approved by the Institutional Review Board at the University of Utah. The following primary human fibroblasts were obtained from the Coriell Cell Repositories (Camden, NJ, USA): ALS patient (TDP-43$^{G298S}$ mutation) (ND32947), Huntington disease (HD) patient (ND33392) and SCA3 (Machado-Joseph disease, MJD) patient (GM06153). TDP-43$^{G298S}$, HD, SCA3, and HEK-293 cells were maintained in DMEM medium containing 10% fetal bovine serum.

For overexpression of recombinant proteins, HEK-293 cells were seeded on 100 mm or 6-well dishes and incubated overnight. The cells were then transfected with plasmid DNAs and harvested 48 h post-transfection and processed as two aliquots for protein and RNA analyses. For siRNA experiments, cells were transfected with siRNAs using lipofectamine 2000 transfection reagent (ThermoFisher Scientific), and lymphoblastoid B cells (1 × 10$^6$) were electroporated with siRNAs using the Neon transfection system (Invitrogen, USA) according to the manufacturer's protocol and seeded on 6-well plates. Prior standardization experiments showed that maximum silencing was achieved 4–5 days post-transfection/electroporation.

**Generation of HEK-293-ATXN2-Q22/58 knock-in (KI) cells**. CRISPR/Cas9-mediated gene editing was carried out according to published protocols[45]. The human ATXN2 locus was engineered to replace CAG22 with CAG58 repeats in ATXN2 exon-1 in a BAC clone (RP11-798L5) containing ATXN2 (Empire Genomics, USA). The modified CAG58 repeat with flanking locus-specific 0.6 Kb left and right homologous arms (HAs) were cloned into a donor vector. The sgRNA oligonucleotides targeting the ATXN2 locus were annealed and cloned into the hSpCas9-2A-Puro (PX459) vector (Addgene, Cat# 62988). HEK-293 cells were transfected with linearized donor vector and single guide RNA pgRNA-Cas9 vector using lipofectamine 2000 transfection reagent. The cells were cultured with 1.0 µg/ml puromycin for ~ 7–10 days. The puromycin-resistant cells were plated and cultured in a 96-well plate at 1 cell/well until most wells were ~ 80% confluent with cells. The cells were then expanded and maintained for PCR screening to identify knock-in positive cells.

**Mice**. ATXN2$^{Q127}$ (Pcp2-ATXN2[Q127]) mice[8] were maintained in a B6D2F1/J background and BAC-SCA2 mice (BAC-Q22 and BAC-Q72) (ref. [9]) in a C57BL/6J and FVB/NJ mixed background, and SCA2 knock-out [Atxn2$^{-/-}$] mouse[49] in a C57BL/6J and 129 × 1/SvJ mixed background were used. The Stau1$^{tm1Apa(-/-)}$ (Stau1$^{-/-}$) mouse[31] was a generous gift from Prof. Michael A. Kiebler, Ludwig Maximilian University of Munich, Germany, and maintained in a C57BL/6J background. Genotyping of animals was accomplished according to published protocols[8,9,31,49]. All mice were bred and maintained under standard conditions consistent with National Institutes of Health guidelines and conformed to an approved University of Utah IACUC protocol.

**Antibodies**. The antibodies used for western blotting and their dilutions were as follows: mouse anti-Ataxin-2 antibody (Clone 22/Ataxin-2) [(1:4000), BD Biosciences, Cat# 611378], rabbit anti-Staufen antibody [(1:5000), Novus biologicals, NBP1-33202], DDX6 antibody [(1:5000), Novus biologicals, NB200-191], RGS8 antibody [(1:5000), Novus Biologicals, NBP2-20153], LC3B Antibody [(1:7000), Novus biologicals, NB100-2220], TDP-43 antibody [(1:7000), Proteintech, Cat# 10782-2-AP], monoclonal anti-FLAG M2 antibody [(1:10,000), Sigma-Aldrich, F3165], monoclonal Anti-Calbindin-D-28K antibody [(1:5000), Sigma-Aldrich, C9848], monoclonal anti-β-Actin−peroxidase antibody (clone AC-15) [(1:20,000), Sigma-Aldrich, A3854], PCP-2 antibody (F-3) [(1:3000), Santa Cruz, sc-137064], Homer-3 antibody (E-6) [(1:2000), Santa Cruz, sc-376155], Anti-PCP4 antibody [(1:5000), Abcam, ab197377], Anti-FAM107B antibody [(1:5000), Abcam, ab175148], rabbit anti-PABP antibody [(1:4000), Abcam, ab153930], p21 Waf1/Cip1 (12D1) rabbit mAb [(1:7000), Cell Signaling, Cat# 2947], SQSTM1/p62 antibody [(1:4000), Cell Signaling, Cat# 5114], Cyclin B1 (V152) mouse mAb [(1:5000), Cell Signaling, Cat# 4135], anti-Polyglutamine-Expansion diseases marker antibody, clone 5TF1-1C2 [(1:3000), EMD Millipore, MAB1574], rabbit anti-neomycin phosphotransferase II (NPTII) antibody [(1:5000), EMD Millipore, AC113], anti-Myc-HRP antibody [(1:5000), Invitrogen, P/N 46-0709], 6 × -His Tag Monoclonal Antibody (HIS.H8), HRP [(1:10,000), ThermoFisher Scientific, MA1-21315-HRP] and sheep-anti-Digoxigenin-POD, Fab fragments [(1:10,000), Roche Life Science, Cat# 11207733910]. The secondary antibodies were: Peroxidase-conjugated horse anti-mouse IgG (H + L) antibody [(1:5000), Vector laboratories, PI-2000] and Peroxidase-conjugated AffiniPure goat anti-rabbit IgG (H + L) antibody [(1:5000), Jackson ImmunoResearch Laboratories, Cat# 111-035-144].

**Immunofluorescence**. Immunofluorescence studies were performed to determine the co-localization of ATXN2 and Staufen1. Briefly, SCA2 FBs and ATXN2-Q22/58 KI cells were plated on cover slides overnight and fixed with 4% paraformaldehyde/PBS. For stress experiments, HEK-293 cells or SCA2 FBs were cultured on cover

slides for overnight and treated with 0.5 mM arsenite (Sigma-Aldrich) for 15 and 30 min at 37 °C or by heat-shocked at 43.5 °C for 1 h. For fluorescence in situ hybridization (FISH) studies, a Cy3-conjugated PCP2 oligonucleotide probe was used to detect STAU1-PCP2 RNA granules in SCA2 FBs according to publish method[66]. Cells were fixed with 4% paraformaldehyde/PBS, permeabilized with 0.1% Triton X-100, and processed for immunostaining using corresponding primary and fluorescent secondary antibodies. The nuclei were stained with DAPI followed by mounting with Fluoromount-G (Southern Biotech, Cat# 0100-01). Paraffin-embedded brain slices from SCA2 patient were provided by Prof. Arnulf H. Koeppen, M.D., Albany Medical College, New York, USA. Non-SCA2 control paraffin-embedded brain slices were provided by Dr. Sonnen, Pathologist, University of Utah. Human tissues were maintained and processed under standard conditions consistent with National Institutes of Health guidelines and conformed to an approved University of Utah IRB protocol. Isolated cerebella were fixed in 4% paraformaldehyde and embedded in paraffin, and then deparaffinized using standard conditions. For cryoprotected cerebella, free-floating sections were used for immunostaining. Sections were blocked/permeabilized with 5% donkey serum 0.3% Triton X-100 in PBS and processed for immunostaining. Images were acquired using confocal microscope (Nikon Eclipse Ti microscopy) in lab or University of Utah cell imaging core lab, and analyzed by Nikon EZ-C1 or NIS-Elements AR 4.5 software. The co-localization plugin in ImageJ was used to define co-localized areas via intensity-based thresholding. The Analyze Particles tool was then used to count the number of co-localized areas greater than four pixels (baseline). The following antibody dilutions were used for cellular immunostainings: mouse anti-Ataxin-2 antibody (Clone 22/Ataxin-2) [(1:750), BD Biosciences, Cat# 611378], rabbit anti-Staufen antibody [(1:750), Novus biologicals, NBP1-33202] and TIA-1 antibody (C-20) [(1:500), Santa Cruz, sc-1751]. Antibody dilutions for tissue immunostainings were custom-designed ATXN2 rabbit polyclonal antibody [SCA2-280 (1:250) (ref. [8]) and STAU1 antibody (C-4) [(1:250), Santa Cruz, sc-390820]. Fluorescent secondary antibodies were: Alexa Fluor® 488-conjugated AffiniPure Donkey Anti-Mouse IgG (H + L) [(1:1000), Jackson ImmunoResearch Laboratories, Cat# 715-545-150], Cy™3-conjugated AffiniPure Donkey Anti-Rabbit IgG (H + L) [(1:1000), Jackson ImmunoResearch Laboratories, Cat# 711-165-152], DyLight™405-conjugated AffiniPure Donkey Anti-Goat IgG (H + L) [(1:1000), Jackson ImmunoResearch Laboratories, Cat# 705-475-147], goat anti-mouse IgG (H + L) antibody, DyLight-488 [(1:1000), ThermoFisher Scientific, Cat# 35502], goat anti-mouse IgG antibody, DyLight-550 [(1:1000), Thermo Scientific, Cat# 84540], goat anti-rabbit IgG (H + L) antibody, DyLight-488 [(1:1000), ThermoFisher Scientific, Cat# 35552] and goat anti-rabbit IgG, DyLight-549 antibody [(1:1000), Thermo Scientific, Cat# 35557].

**Preparation of protein lysates and western blot analyses**. Cellular extracts were prepared by a single-step lysis method[9,42]. The harvested cells were suspended in SDS–PAGE sample buffer [Laemmli sample buffer (Bio-Rad, Cat# 161-0737]) and then boiled for 5 min. Equal amounts of the extracts were used for western blot analyses. Cerebellar protein extracts were prepared by homogenization of mouse cerebella in extraction buffer [25 mM Tris-HCl pH 7.6, 300 mM NaCl, 0.5% Nonidet P-40, 2 mM EDTA, 2 mM MgCl$_2$, 0.5 M urea and protease inhibitors (Sigma-Aldrich, P8340)] followed by centrifugation at 4 °C for 20 min at 14,000 RPM. Only supernatants were used for western blotting. Protein extracts were resolved by SDS–PAGE and transferred to Hybond P membranes (Amersham Bioscience, USA). After blocking with 4% skim milk in 0.1% Tween 20/PBS, the membranes were processed for immunostaining using corresponding primary and secondary antibodies. Signals were detected by using the Immobilon Western Chemiluminescent HRP Substrate (EMD Millipore, WBKLS0500) according to the manufacturer's protocol. Blot images were scanned and band intensities were quantified by ImageJ software analyses after inversion of the images, and the relative protein abundances were expressed as ratios to β-Actin. Detailed western blots for main figures in text are presented in Supplementary Figs. 6–14.

**Immunoprecipitations**. To determine protein–protein or protein–RNA interactions, we carried out protein–RNA immunoprecipitation (IP) experiments using HEK-293 cells expressing Flag-ATXN2-Q22 and Flag-ATXN2-Q108 or Flag-STAU1. The preparation of whole-cell extracts and IP methods followed previously published methods[9,42]. First, cells were lysed with a cytoplasmic extraction buffer [25 mM Tris-HCl pH 7.6, 10 mM NaCl, 0.5% NP40, 2 mM EDTA, 2 mM MgCl$_2$ and protease inhibitors] and cytoplasmic extracts were separated by centrifugation at 14,000 RPM for 20 min. Second, the resultant pellets were suspended in nuclear lysis buffer or high salt lysis buffer (25 mM Tris-HCl, pH 7.6, 500 mM NaCl, 0.5% Nonidet P-40, 2 mM EDTA, 2 mM MgCl$_2$ and protease inhibitors), and the nuclear extracts were separated by centrifugation at 14,000 RPM for 20 min. The nuclear extracts were then combined with the cytoplasmic extracts and denoted whole-cell extracts. Specifically, while combining cytoplasmic and nuclear extracts, the NaCl concentration was adjusted to physiologic buffer conditions (~ 150 mM) to preserve in vivo interactions. For identifying non-RNA mediated interactions, whole-cell extracts were treated with 1.0 mg/ml RNase A (Amersham Bioscience, USA) for 15 min at 37 °C before subjected to IP. Ninety percent of cell extracts were subjected to Flag monoclonal antibody (mAb) IP (Anti-Flag M2 Affinity Gel, Sigma-Aldrich, A2220-1ML) to immunoprecipitate ATXN2 or STAU1 interacting protein–RNA complexes. The remaining 10% of whole-cell extracts were saved as

the input control for western blotting and RT-PCR analyses. The IPs were washed with a buffer containing 200 mM NaCl and the bound protein–protein or protein–RNA complexes were eluted from the beads with Flag peptide competition (100 μg/ml) (Flag Peptide, Sigma-Aldrich, F3290-4MG). Eluted fractions were divided into two equal parts. One part was analyzed by SDS–PAGE followed by western blotting to determine interactions between ATXN2 and STAU1. RNA was isolated from the other fraction and subjected to RT-PCR analyses to identify RNAs that bound to STAU1.

**In vitro RNA-binding (Northwestern) assay**. Northwestern blot assays were performed following a published protocol with some modifications[67]. BL21 < DE3 > cells carrying His-STAU1, His-GFP and empty pET vectors, were grown to mid-log phase. Whole-cell lysates from harvested cells were run on SDS–PAGE and electro-blotted onto a Hybond P membrane. The transferred proteins were re-natured as follows: the blot was first incubated with binding buffer (0.1 M HEPES, pH 7.9, 0.1 M $MgCl_2$, 0.1 M KCl and 0.5 μM $ZnSO_4$) with 1 mM dithiothreitol (DTT) and 6 M urea for 5 min at room temperature. The blots were then incubated, for 5 min each, through five serial twofold dilutions of urea with binding buffer/1 mM DTT with continuing incubation steps until the binding buffer was 1 mM DTT without urea. Blot pre-hybridizations were carried out with binding buffer (1 mM DTT, 5% bovine serum albumin (BSA) and 1 μg/ml yeast tRNA) for 1 h at room temperature. The blots were then hybridized with DIG-labeled RNA probes in binding buffer (1 mM DTT, 0.25% BSA and 1 μg/ml yeast tRNA) overnight at 4 °C. After several washes with 0.1% Tween 20/PBS, the membranes were incubated with anti-DIG-POD antibody in 4% skim milk in 0.1% Tween 20/PBS for 2 h at room temperature. Following three additional washes with 0.1% Tween 20/PBS, signals were detected by using the Immobilon Western Chemiluminescent HRP Substrate according to the manufacturer's protocol.

**RNA expression analyses by quantitative RT-PCR**. Mice were deeply anesthetized with isoflurane. Mouse cerebella were removed and immediately submerged in liquid nitrogen. Tissues were kept at −80 °C until the time of processing. Total RNA was extracted from mouse cerebella and from harvested cells using the RNaeasy mini-kit according to the manufacturer's protocol (Qiagen, USA). DNAse I treated RNAs were used to synthesize cDNAs using the ProtoScript cDNA synthesis kit (New England Biolabs, USA). Quantitative RT-PCR was performed in QuantStudio 12 K (Life Technologies, USA) with the Power SYBR Green PCRMaster Mix (Applied Biosystems, USA) using University of Utah genomics core lab. PCR reaction mixtures contained SYBR Green PCRMaster mix, synthesized cDNA and 0.5 pmol primers, and PCR amplifications were carried out for 45 cycles: denaturation at 95 °C for 10 s, annealing at 60 °C for 10 s and extension at 72 °C for 40 s. The threshold cycle for each sample was chosen from the linear range and converted to a starting quantity by interpolation from a standard curve run on the same plate for each set of primers. All gene expression levels were normalized to *ACTB* or *GAPDH* or *Actb* mRNA levels. Primer pairs designed for qRT-PCR and RT-PCR are given as forward and reverse, respectively, and listed in Supplementary Table 2.

**Rotarod testing**. Following genotyping mice were randomly assigned to cages according to sex, ensuring that each possible genotype was represented in each cage. During the five days of testing, mice were taken to a separate testing room and allowed to habituate for 1 h. Testing began at the same time every day and was completed by the same technician. The technician was blinded to the genotypes of the mice in each cage. On day 1 mice were handled for 2 min per mouse. On day 2 mice were placed on an accelerating rotarod apparatus (Rotamex-5, Columbus Instruments, Columbus, OH, USA) initially rotating at 4 RPM for 2 min. The rate of rotation was increased by 1 RPM per every 15 s to 10 RPM for 60 s. On days 3–5, mice were tested on the accelerating rotarod from 0 RPM with acceleration increasing by 1 RPM every 9 s until the maximum speed of 51 RPM was reached (maximum time of 459 s). The time that a mouse fell (latency to fall) from the rotating bar was recorded. Statistical comparisons of rotarod data were determined using the method of generalized estimating equations (GEE) with the independent correlation option using Stata 12 (procedures xtset followed by xtgee).

**Statistical analysis**. Student's $t$-tests and analysis of variance (ANOVA) were used to determine whether differences between groups were significant. The level of significance was set at $P \leq 0.05$. *$P \leq 0.05$, **$P \leq 0.01$, ***$P \leq 0.001$ and ns = $P >$ 0.05. Means ± SD are presented throughout, unless otherwise specified.

## Data availability

The data that support the findings of this study are included in this published article and its Supplementary Information.

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

## Acknowledgements

We thank Duong P. Huynh, Ph.D. for providing *ATXN2* clones with expanded CAG repeats and valuable discussion throughout this study. We thank Prof. Dr. Michael A. Kiebler, Ludwig Maximilian University of Munich, Germany for providing *Stau1$^{tm1Apa(-/-)}$* (*Stau1$^{-/-}$*) mice. We thank Matthew Schneider, Heydon Kaddas and Christopher Nelson for technical assistance and mouse handlings. We thank Mandi Gandelman for instruction of mouse tissue sectioning, Erika Aoyama for part of cellular immunos-taining, and also thank Amber Staples and Micah Caleb Tyler for immunohistological image analyses. This work was supported by grants R01NS097903, RC4NS073009, and R56NS33123 from the National Institutes of Neurological Disorders and Stroke to S.M.P., and Noorda foundation to S.M.P., and grants R37NS033123 and R21NS081182 to D.R.S. and S.M.P. S.M.P. received grant support from the Target ALS Foundation and is a consultant for Ataxion Pharmaceuticals and Progenitor Life Sciences. The funders had no role in study design, data collection and analysis, decision to publish, or preparation of the manuscript.

## Author contributions

S.P., W.D. and S.M.P. conceived and designed the experiments. Experiments were performed by S.P. and W.D. Luciferase experiments were conducted by D.R.S. Data analyses was performed by S.P., W.D., D.R.S. and S.M.P. All figures were generated by S.P. and W.D. Administrative assistance was provided by K.P.F. The manuscript was written by S.P., W.D., D.R.S. and S.M.P., and critically reviewed by D.R.S. and S.M.P.

## Additional information

**Competing interests:** The authors declare no competing interests.

