## [Peer Review File · Nature Communications]

Reviewers' comments:

Reviewer #1 (Remarks to the Author):

This work by the laboratories of Drs. Scoles and Pulst provides new, interesting and novel information on the biology of disease in the polyQ disease SCA2, and highlights a new potential target for therapy in this age-related and currently incurable disease.

SCA2, like the other members of the polyQ family of diseases, is caused by anomalous expansion in the CAG triplet repeat in the gene ATXN2. This translates into an abnormally long polyQ repeat in the RNA-functioning protein, ATXN2. The data provided by the authors make a convincing case of an increased interaction of SCA2-causing ATXN2 with the stress granule protein Staufen-1. Based on a series of cell-oriented and, at the end, animal-based studies, the authors conclude that expansion in the polyQ region of atxn2 leads to stabilization of staufen-1, which, in turn, disrupts the expression of a handful of genes. Targeting Staufen-1 in SCA2 mice ameliorates anomalies and improves pathology in SCA2-modeling mice.

Overall the work is executed well and the conclusions are reached appropriately. There are a few issues that arose for me as I read through the work.

Major concerns:

- There is a lack of mechanistic information on how expanded ATXN2 regulates the stability of staufen-1. Is it through the proteasome pathway? through autophagy? By changing its interactions with other proteins? A possible role in the protein stability pathways would be an interesting finding for atxn2, as it might increase what we know about this protein and may reach a wider audience for this manuscript.
- For a journal such as Nature Communications, I am a little concerned that the impact of these interesting findings might not be wide enough; they are rather limited in scope to SCA2. In the absence of any additional data, perhaps an expansion of the Discussion section, and additional perspective provided in the Discussion can go a long way.

Minor issues:

- I wonder if some of the work in SCA2 cell model might not benefit from studies using another SG protein, one that is not staufen and which does not interact with atxn2 to further cement the specificity of staufen-dependent effects in SCA2.
- I think that all of the supplementary data should be included in the main figures.

Reviewer #2 (Remarks to the Author):

This manuscript describes a novel interaction between ATXN2, the disease protein underlying SCA2 and the RNA-binding protein STAU1. The authors show that these two proteins co-localize in stress granules (SGs) and then go on to show that Staufen1 protein but not RNA expression is increased in SCA2 patient cell lines and mouse models. STAU1 is shown to bind the 3' UTR of PCP2 mRNA and Staufen1 expression is shown to be inversely correlated with PCP2 mRNA levels. Finally, the authors show that Staufen gene dose reduction by 50% improves motor behavior of ATXNQ127 transgenic mice. While these findings are interesting, unfortunately the results are presented in a confusing and incomplete way. The manuscript lacks a consistent presentation of the findings in one relevant system. Moreover, the underlying mechanism remains unclear. The authors must be aware of these shortcomings given their choice of a rather non-descriptive title of the manuscript.

Specific comments:

1) The co-localisation of STAU1 and ATXN2 in some of the induced SGs is interesting. However, the relevance of this finding to the rest of the manuscript is unclear. Are any of the subsequently shown findings related to stress granules? For example, does ATXN2 recruit STAU1 to SGs or vice versa? Is ATXN2 still present in SGs upon STAU1 knockdown? In addition, is PCP2 mRNA also found in stress granules? It would also be important to know whether STAU1 is present in SG in ATXN2 aggregates in SCA2 brains in both mouse models and human.

2) Fig. 1g: The levels of STAU1 appear unchanged upon overexpression of ATXN2-Q108. This is in stark contrast to the findings shown in Figures 2 and 3 and the point that the authors are trying to make later and needs to be addressed by the authors. Could the authors also comment on the presence of 2 bands in the STAU1 WB?

3) Supplementary Fig1: It would be important to show specificity of the anti-Staufen antibody not only in WB analysis but also for immunostaining, particularly the shown immunostaining in Fig. S1A, which is unclear.

4) Figure 2: What about RNA expression of STAU1; is this also increased in the patient cell lines and mouse cerebellar extracts?

5) Figure 2C would be better shown in Supplementary Information linked to Figure 4A. I also think that the presentation of the findings in panel E,F and G is confusing at this point. The authors should discuss these findings later in the context of the findings shown in Figure 6E-G. What would be the mechanism of why STAU1 is reduced only upon silencing ATXN2 in the context of ATXN-52 overexpression but not of ATXN2 wildtype expression? However, as so often in this manuscript, these experiments are done in different cell lines so the difference could also be due to the difference in cell type?

6) Figure 3A: Please see my comment above. Why is STAU1 expression increased here but not in Fig1G? Also, the authors later do not comment at all later in the manuscript on the potential post-transcriptional mechanism by which ATXN2 might influence STAU1 protein levels.

7) Figure 4A: What about STAU1 mRNA levels in the patient LB cells? And are PCP2 protein levels reduced? It would be good to show both qPCR and WB for all proteins involved in the lymphoblastoid cells.

8) Figure 4B: Are protein levels of PCP2 and CALB1 reduced?

9) Page 10: Please note that the authors have only shown an effect on PCP2 RNA, so referencing Figures 2-4 to make the point that increased abundance of STAU1 in SCA2 cells is associated with a general dysregulation of RNA processing is incorrect.

10) Figure 6A-D: As before, it would be good to have a complete picture and show WB results for all the proteins involved. The authors should include panels on PCP2 and ATXN2.

11) Figure 6H-J could go into supplementary information as it shows the same results as shown in Figure 6E-G.

12) Figure 6K is a key figure of the paper and that unfortunately leaves many questions open. It would be important to show actual STAU1 protein levels in the cerebella of these mice together with ATXN2, PCP2, and even CALB protein levels to bring the other results of the paper together in one relevant system. And linking it back to Figure 1, what about stress granules in the brains in these mice? Finally, I am unclear about the exact genetic backgrounds of the different mouse strains used. The authors should provide more details in their Methods. Can the authors confirm that the ATXN2Q127 mice are on exactly the same B6 background as the Stau1 mice? If not, an

essential control would be missing, i.e., a cross of the ATXN2Q127 with wildtype mice of the Stau1 mice background as the genetic cross per se might have behavioural consequences.

13) The authors should discuss the proposed molecular mechanism in their Discussion. Perhaps add a figure on that?

Minor comments:

- The manuscript would benefit from proofreading as grammar and style could be improved in several places.
- The authors might want to say “We demonstrated elevated abundance of the RNA-binding protein Staufen1 (STAU1) in SCA2 fibroblasts...” for clarity
- The authors should introduce what patient LB cells are on page 6.

Reviewer #3 (Remarks to the Author):

Spinocerebellar ataxia type 2 (SCA2) is a rare neurodegenerative disorder due to a CAG repeat expansion within the ataxin-2 gene. Aggregation of the mutant protein contributes to altered RNA metabolism which leads to pathological consequences. Treatment of the disorder is primarily palliative with treatment for the major symptoms which is not curative. The present submission addresses mechanism and goes some way to assisting in our understanding as to abnormalities in RNA metabolism when ATXN2 is mutated by repeat expansion. It has been established that ATXN2 localises to stress granules but its function there remains unknown.

This research is novel in that it reveals a relationship between ATXN2 and Staufen 1 (STAU1) a protein involved in RNA decay. Stau1 is elevated in SCA2 fibroblasts and an association was observed with decreased levels of PCP2 and CALB1 mRNA and protein. Silencing of Stau1 restored levels of PCP2. This control appeared to be due to binding of Stau1 to the 3UTR of PCP. Silencing of ATXN2 also lead to normal levels of Stau1 and PCP and reduction of Stau1 gene dosage improved motor behaviour in SCA2 mice.

This work is of considerable interest to researchers with an interest in neurodegeneration and in more general to the field of RNA binding proteins. The quality of the data is generally very good and thorough with well controlled experiments.

Specific points:

1. It is evident that Stau1 stains different cell types in the cerebellum. This could be made clearer by co-staining with calbindin since some of these cells appear to be Purkinje neurons.
2. Why was p21 elevated at zero time in Fig 3b?
3. What is the significance of localisation of ATXN2 and Staufen 1 to stress granules? How does it fit into the novel function of Staufen 1 in RNA metabolism?
4. Again on that point is there evidence for re-initiation of protein translation in SCA2 cells under stress?

Reviewer #1 (Remarks to the Author):

This work by the laboratories of Drs. Scoles and Pulst provides new, interesting and novel information on the biology of disease in the polyQ disease SCA2, and highlights a new potential target for therapy in this age-related and currently incurable disease.

SCA2, like the other members of the polyQ family of diseases, is caused by anomalous expansion in the CAG triplet repeat in the gene ATXN2. This translates into an abnormally long polyQ repeat in the RNA-functioning protein, ATXN2. The data provided by the authors make a convincing case of an increased interaction of SCA2-causing ATXN2 with the stress granule protein Staufen-1. Based on a series of cell-oriented and, at the end, animal-based studies, the authors conclude that expansion in the polyQ region of atxn2 leads to stabilization of staufen-1, which, in turn, disrupts the expression of a handful of genes. Targeting Staufen-1 in SCA2 mice ameliorates anomalies and improves pathology in SCA2-modeling mice.

Overall the work is executed well and the conclusions are reached appropriately. There are a few issues that arose for me as I read through the work.

We appreciate the overall very positive comments.

Major concerns:

There is a lack of mechanistic information on how expanded ATXN2 regulates the stability of staufen-1. Is it through the proteasome pathway? through autophagy? By changing its interactions with other proteins? A possible role in the protein stability pathways would be an interesting finding for atxn2, as it might increase what we know about this protein and may reach a wider audience for this manuscript.

We now show that SCA2 cells have abnormal autophagy; cells expressing polyQ expanded ATXN2 have increased LC3-II and p62 as markers of disturbed autophagy (Fig. 3f-h). On the other hand, treating these cells with proteasome inhibitor had no effect on STAU1 levels (Fig. 3d-e). The finding is consistent with impaired autophagy seen in other polyQ cellular models, now referenced in the Discussion.

For a journal such as Nature Communications, I am a little concerned that the impact of these interesting findings might not be wide enough; they are rather limited in scope to SCA2. In the absence of any additional data, perhaps an expansion of the Discussion section, and additional perspective provided in the Discussion can go a long way.

We performed additional experiments to show that STAU1 protein abundance is increased in ALS-patient FBs harboring the TDP-43^{G298S} mutation. We also replicated increased STAU1 levels in cultured cells overexpressing wild-type TDP-43, and mutant TDP-43^{G298S} (Figs. 2e & S2e). Of note, wildtype TDP-43 is found in neuronal aggregates in the majority of ALS patients, even those with mutations in other ALS genes.

Minor issues:

I wonder if some of the work in SCA2 cell model might not benefit from studies using another SG protein, one that is not staufen and which does not interact with atxn2 to further cement the specificity of staufen-dependent effects in SCA2.

We would like to clarify that we had already demonstrated that expression of the SG protein DDX6, which interacts with ATXN2, was not changed by ATXN2 mutation (Fig. 2a,b and Dansithong et al., 2015). We now also show that the classic SG marker protein TIA1 co-localizes in granules with STAU1 and ATXN2 (Fig. 1a).

I think that all of the supplementary data should be included in the main figures.

In the revised manuscript we have reorganized the presentation of data including moving some of the supplementary data to the main figures.

Reviewer #2 (Remarks to the Author):

This manuscript describes a novel interaction between ATXN2, the disease protein underlying SCA2 and the RNA-binding protein STAU1. The authors show that these two proteins co-localize in stress granules (SGs) and then go on to show that Staufen1 protein but not RNA expression is increased in SCA2 patient cell lines and mouse models. STAU1 is shown to bind the 3' UTR of PCP2 mRNA and Staufen1 expression is shown to be inversely correlated with PCP2 mRNA levels. Finally, the authors show that Staufen gene dose reduction by 50% improves motor behavior of ATXN2 transgenic mice. While these findings are interesting, unfortunately the results are presented in a confusing and incomplete way. The manuscript lacks a consistent presentation of the findings in one relevant system. Moreover, the underlying mechanism remains unclear. The authors must be aware of these shortcomings given their choice of a rather non-descriptive title of the manuscript.

Please see above (reviewer 1) regarding mechanistic insights into STAU1 protein elevation. We have reorganized the figures and changed the manuscript title. Whenever possible, observations were verified in multiple cell lines, now also including CRISPR/Cas9-modified HEK-293^{ATXN2Q22/Q58} knockin cells and their isogenic HEK-293 controls.

Specific comments:

1) The co-localisation of STAU1 and ATXN2 in some of the induced SGs is interesting. However, the relevance of this finding to the rest of the manuscript is unclear. Are any of the subsequently shown findings related to stress granules? For example, does ATXN2 recruit STAU1 to SGs or vice versa? Is ATXN2 still present in SGs upon STAU1 knockdown? In addition, is PCP2 mRNA also found in stress granules? It would also be important to know whether STAU1 is present in SG in ATXN2 aggregates in SCA2 brains in both mouse models and human.

We now show that ATXN2-positive SGs are greatly reduced with *STAU1* haploinsufficiency (Fig. 7d) and that *PCP2* mRNA colocalizes with STAU1-positive granules (Fig. 5d). STAU1 is present in mutant ATXN2 aggregates in mouse and man (Fig. 1e, f). These results tie together the morphological characteristics of stress (granules), the importance of STAU1 and its interaction with ATXN2 in these granules, and the interaction with specific key mRNAs (*Pcp2*). Also note the Venn diagram in Supp Fig. 5 that shows that many differentially regulated mRNAs in SCA2 cerebellum are direct STAU1 targets as identified by hiCLIP.

2) Fig. 1g: The levels of STAU1 appear unchanged upon overexpression of ATXN2-Q108. This is in stark contrast to the findings shown in Figures 2 and 3 and the point that the authors are trying to make later and needs to be addressed by the authors. Could the authors also comment on the presence of 2 bands in the STAU1 WB?

The blot in Fig. 1g examines RNase dependence of the STAU1/ATXN2 interaction and was not controlled/normalized for STAU1 input levels. We have now replicated this study controlling for STAU1 input (Fig. 1g). Regarding the two bands, note that the Staufen antibody detects multiple isoforms of Staufen1 including a low molecular weight form, consistent with information in the NCBI database. We have made reference to this in the revised manuscript.

3) Supplementary Fig1: It would be important to show specificity of the anti-Staufen antibody not only in WB analysis but also for immunostaining, particularly the shown immunostaining in Fig. S1A, which is unclear.

We show specificity of the Staufen antibody using Staufen1 RNAi in human HEK-293 cells, mouse Neuro 2A cells, and mouse cerebellar extracts by western blot analyses (Supp Fig. 1a-c), and in cells by immunocytochemistry (Supp Fig. 1d).

4) Figure 2: What about RNA expression of STAU1; is this also increased in the patient cell lines and mouse cerebellar extracts?

STAU1 RNA levels are not changed in SCA2 patient cell lines and mouse cerebellar extracts. These data are included in **Fig. 2 (f-h)**.

5) Figure 2C would be better shown in Supplementary Information linked to Figure 4A. I also think that the presentation of the findings in panel E,F and G is confusing at this point. The authors should discuss these findings later in the context of the findings shown in Figure 6E-G. What would be the mechanism of why *STAU1* is reduced only upon silencing *ATXN2* in the context of *ATXN2* overexpression but not of *ATXN2* wildtype expression? However, as so often in this manuscript, these experiments are done in different cell lines so the difference could also be due to the difference in cell type?

Please note that figures have been reorganized. At this point, the answer regarding lack of *STAU1* knockdown effect in wildtype cells has to remain speculative, although the observation itself is confirmed *in vitro* (**Supp Fig. 4**) and *in vivo* (**Fig. 7b**). It is possible that the action of normal *ATXN2* on *STAU1* (and downstream mRNA targets) requires the presence of stress and stress granules. In other words, the formation of stress granules in association with *STAU1* increase may be necessary to engage *STAU1*-mediated mRNA decay. This is supported by a recent study of autophagy in *SOD1* mice that showed significant spinal cord transcriptomic changes caused by knocking out autophagy in *SOD1* mice but when done in wildtype mice few changes were observed.

6) Figure 3A: Please see my comment above. Why is *STAU1* expression increased here but not in Fig1G? Also, the authors later do not comment at all later in the manuscript on the potential post-transcriptional mechanism by which *ATXN2* might influence *STAU1* protein levels.

Please see answer to comment 2. Regarding *ATXN2* influence on *STAU1* protein levels, please see answers to R1 above.

7) Figure 4A: What about *STAU1* mRNA levels in the patient LB cells? And are *PCP2* protein levels reduced? It would be good to show both qPCR and WB for all proteins involved in the lymphoblastoid cells.

STAU1 RNA levels are not changed in SCA2 patient LBCs (**Fig. 2g**).

In FBs and LBCs, *PCP2* mRNAs are present but not translated at detectable levels. *PCP2* mRNA levels are reduced in both cell types in the presence of mutant *ATXN2*. In HEK-293 cells, however, *PCP2* protein is detectable by western blot analysis and is reduced by exogenous expression of *STAU1* (see **Fig. 4c** for protein, **4d** for mRNA).

8) Figure 4B: Are protein levels of *PCP2* and *CALB1* reduced?

In HEK-293 cells, *PCP2* protein but not *CALB1* is detectable by western blot analysis and is reduced by exogenous expression of *STAU1* (see **Fig. 4c** for *PCP2* protein, **4d** for *PCP2* and *CALB1* mRNAs).

9) Page 10: Please note that the authors have only shown an effect on *PCP2* RNA, so referencing Figures 2-4 to make the point that increased abundance of *STAU1* in SCA2 cells is associated with a general dysregulation of RNA processing is incorrect.

The reviewer is correct. We have only shown direct interaction of *STAU1* with the 3' UTR of *PCP2*. We have now clearly stated that extension to a more global mRNA dysregulation is speculative, though plausible given the recognized role of *STAU1* binding to 3' UTRs of mRNAs. More importantly, hiCLIP data in HEK293 cells from the Ule group (Sugimoto et al., 2015) show a surprising overlap of *STAU1*-regulated transcripts with our *in vivo* mouse transcriptomes (**new Supp Fig. 5**) in that 176 mRNAs that are direct *STAU1* interactors, belong to the set of significantly differentially regulated mRNAs in SCA2 mouse models. This number represents a minimum estimate as many cerebellar transcripts are not abundantly expressed in HEK-293 cells (for example *RGS8*).

10) Figure 6A-D: As before, it would be good to have a complete picture and show WB results for all the proteins involved. The authors should include panels on PCP2 and ATXN2.

We now provide western blots showing PCP2 alterations as a function of STAU1 and ATXN2 status (Fig. 6). Please note that experiments with STAU1-siRNA are replicated in LB and ATXN2-Q22/58 knockin cells (Fig. 6 and Supp Fig. 4). The new Fig. 7b shows *in vivo* protein levels of these and other key proteins as a function of *Stau1* haploinsufficiency.

11) Figure 6H-J could go into supplementary information as it shows the same results as shown in Figure 6E-G.

In the revised manuscript we have reorganized the presentation of data.

12) Figure 6K is a key figure of the paper and that unfortunately leaves many questions open. It would be important to show actual STAU1 protein levels in the cerebella of these mice together with ATXN2, PCP2, and even CALB protein levels to bring the other results of the paper together in one relevant system. And linking it back to Figure 1, what about stress granules in the brains in these mice? Finally, I am unclear about the exact genetic backgrounds of the different mouse strains used. The authors should provide more details in their Methods. Can the authors confirm that the ATXN2Q127 mice are on exactly the same B6 background as the *Stau1* mice? If not, an essential control would be missing, i.e., a cross of the ATXN2Q127 with wildtype mice of the *Stau1* mice background as the genetic cross per se might have behavioural consequences.

Old Fig. 6K is now Fig. 7a. Please see answers above (Fig. 7b) and note the reduction in granules *in vivo* upon STAU1 reduction (Fig. 7d).

The reviewer is correct that mouse background can have important effects on phenotypes. We have therefore used littermates to compare behavioral effects in different genotypes. Thus, the 4 different genotypes shown in Fig. 7 all occur on the same mixed B6:D2 hybrid background. The results also show conclusively that *STAU1* haploinsufficiency by itself does not affect motor performance as the blue (WT) and orange (*Stau1*^{+/-}) tracings virtually overlap. Parenthetically, we want to emphasize that phenotypes observed in hybrid mice –if well controlled by using littermates– are generally more robust and generalizable than those obtained in highly inbred lines.

13) The authors should discuss the proposed molecular mechanism in their Discussion. Perhaps add a figure on that?

In response to the reviewer's comment we provided a figure outlining our working model (Fig. 7e).

Minor comments:

- **The manuscript would benefit from proofreading as grammar and style could be improved in several places.**

We thank the reviewer for the comment and have made extensive effort to improve the grammar of the revised manuscript.

- **The authors might want to say “We demonstrated elevated abundance of the RNA-binding protein Staufen1 (STAU1) in SCA2 fibroblasts...” for clarity.**

We have made the change as recommended in the revised manuscript.

- **The authors should introduce what patient LB cells are on page 6.**

In response to the reviewer's concern we describe SCA2 LBCs in the main text.

Reviewer #3 (Remarks to the Author):

Spinocerebellar ataxia type 2 (SCA2) is a rare neurodegenerative disorder due to a CAG repeat expansion within the ataxin-2 gene. Aggregation of the mutant protein contributes to altered

RNA metabolism which leads to pathological consequences. Treatment of the disorder is primarily palliative with treatment for the major symptoms which is not curative. The present submission addresses mechanism and goes some way to assisting in our understanding as to abnormalities in RNA metabolism when ATXN2 is mutated by repeat expansion. It has been established that ATXN2 localises to stress granules but its function there remains unknown. This research is novel in that it reveals a relationship between ATXN2 and Stau1 (STAU1) a protein involved in RNA decay. Stau1 is elevated in SCA2 fibroblasts and an association was observed with decreased levels of PCP2 and CALB1 mRNA and protein. Silencing of Stau1 restored levels of PCP2. This control appeared to be due to binding of Stau1 to the 3'UTR of PCP. Silencing of ATXN2 also led to normal levels of Stau1 and PCP and reduction of Stau1 gene dosage improved motor behaviour in SCA2 mice.

This work is of considerable interest to researchers with an interest in neurodegeneration and in more general to the field of RNA binding proteins. The quality of the data is generally very good and thorough with well controlled experiments.

We appreciate the positive comments and recognition of the novelty of our findings.

Specific points:

1. It is evident that Stau1 stains different cell types in the cerebellum. This could be made clearer by co-staining with calbindin since some of these cells appear to be Purkinje neurons.

Our calbindin antibody is not compatible with double-labeling for ATXN2 and STAU1. We believe, however, that the morphology of cells in **Fig. 7d** is sufficient to identify them as PCs.

2. Why was p21 elevated at zero time in Fig 3b?

We used p21 (WAF1) as a control owing to the fact that it has a very rapid half-life, thus the marked decrease from 0 to 4 hrs.

3. What is the significance of localisation of ATXN2 and Stau1 to stress granules? How does it fit into the novel function of Stau1 in RNA metabolism?

Our addition of experiments showing that both the stress granule and the protein phenotypes improve in our SCA2 mouse model upon STAU1 knockdown suggest that granule formation and SMD are linked. Clearly, more experiments are needed to separate the roles of mRNA stability, translation, and subcellular localization and formation of granules.

4. Again on that point is there evidence for re-initiation of protein translation in SCA2 cells under stress?

Current dogma would hold that, under stress, translation of key mRNAs is stalled and/or RNAs are aggregated in SGs. We do not know directly whether stalled mRNAs could re-initiate translation upon reduction of STAU1. We do know, however, that reducing STAU1 from elevated levels does increase levels of key proteins *in vitro* and *in vivo*. This can occur either by re-initiation of translation or by reducing SMD (**Fig. 7b**).

Best regards

Stefan M. Pulst, M.D., Dr. med.
Chair and Professor of Neurology
E-mail: stefan.pulst@hsc.utah.edu

REVIEWERS' COMMENTS:

Reviewer #1 (Remarks to the Author):

The authors have sufficiently and satisfactorily addressed my original concerns in this revised form of their manuscript.

Reviewer #2 (Remarks to the Author):

This manuscript describes an interesting novel interaction between ATXN2 and Staufen that might be relevant to the pathogenesis of SCA2 and other neurodegenerative disorders. The revised manuscript has been much improved, and the authors have thoroughly addressed the reviewers' concerns. The overall organization of the manuscript is much improved.

Overall, the work is convincing and opens novel avenues in the field of neurodegenerative diseases. Statistical analyses seem appropriate.

The manuscript would benefit from further language and grammatical proofreading.

Reviewer #3 (Remarks to the Author):

The authors have addressed all the issues I raised to my satisfaction including the recognition of Purkinje cells in Fig 7.

Reviewer #1 (Remarks to the Author):

The authors have sufficiently and satisfactorily addressed my original concerns in this revised form of their manuscript.

We appreciate the overall very positive comment.

Reviewer #2 (Remarks to the Author):

This manuscript describes an interesting novel interaction between ATXN2 and Staufen that might be relevant to the pathogenesis of SCA2 and other neurodegenerative disorders. The revised manuscript has been much improved, and the authors have thoroughly addressed the reviewers' concerns. The overall organization of the manuscript is much improved.

Overall, the work is convincing and opens novel avenues in the field of neurodegenerative diseases. Statistical analyses seem appropriate.

The manuscript would benefit from further language and grammatical proofreading.

We appreciate the positive comments and recognition of the novelty of our findings.

We have made extra effort to improve the grammar of the revised manuscript.

Reviewer #3 (Remarks to the Author):

The authors have addressed all the issues I raised to my satisfaction including the recognition of Purkinje cells in Fig 7.

We appreciate the overall very positive comment.